# Pooled Error Variance and Covariance Estimation of Sparse In Situ Soil Moisture Sensor Measurements in Agricultural Fields in Flanders

Marit G.A. Hendrickx[*,1,2], Jan Vanderborght[1,3], Pieter Janssens[1,4,5], Sander Bombeke[6], Evi Matthyssen[7], Anne Waverijn[8], Jan Diels[1,2]

[1] Department of Earth and Environmental Sciences, KU Leuven, Leuven, 3001, Belgium
[2] KU Leuven Plant Institute (LPI), KU Leuven, Leuven, 3001, Belgium
[3] Agrosphere Institute IBG-3, Forschungszentrum Jülich GmbH, Jülich, 52425, Germany
[4] Soil Service of Belgium, Leuven, 3001, Belgium
[5] Department of Biosystems, KU Leuven, Leuven, 3001, Belgium
[6] Proefstation voor de Groenteteelt, Sint-Katelijne-Waver, 2860, Belgium
[7] Praktijkpunt Landbouw Vlaams-Brabant, Herent, 3020, Belgium
[8] Viaverda vzw, Kruishoutem, 9770, Belgium

*Correspondence to*: Marit Hendrickx (marit.hendrickx@kuleuven.be)

**Abstract.** Accurately quantifying errors in soil moisture measurements from in situ sensors at fixed locations is essential for reliable state and parameter estimation in probabilistic soil hydrological modeling. This quantification becomes particularly challenging when the number of sensors per field or measurement zone (MZ) is limited. When direct calculation of errors from sensor data in a certain MZ is not feasible, we propose to pool systematic and random errors of soil moisture measurements for a specific measurement setup and derive a pooled error covariance matrix that applies to this setup across different fields and soil types. In this study, a pooled error covariance matrix was derived using soil moisture sensor measurements from three TEROS 10 (Meter Group, Inc., USA) sensors per MZ and soil moisture sampling campaigns conducted over three growing seasons, covering 93 cropping cycles in agricultural fields with diverse soil textures in Belgium. The MZ soil moisture estimated from a composite of 9 soil samples with a small standard error (0.0038 $m^3$ $m^{-3}$) was considered the 'true' MZ soil moisture. Based on these measurement data, we established a pooled linear recalibration of the TEROS 10 manufacturer's sensor calibration function. Then, for each individual sensor as well as for each MZ, we identified systematic offsets and temporally varying residual deviations between the calibrated sensor data and sampling data. Sensor deviations from the 'true' MZ soil moisture were defined as observational errors and lump both measurement errors and representational errors. Since a systematic offset persists over time, it contributes to the temporal covariance of sensor observational errors. Therefore, we estimated the temporal covariance of observational errors of the individual and the MZ-averaged sensor measurements from the variance of the systematic offsets across all sensors and MZs averages, while the random error variance was derived from the variance of the pooled residual deviations. The total error variance was then obtained as the sum of these two components. Due to spatial soil moisture correlation, the variance and temporal covariance of MZ-averaged sensor observational errors could not be derived accurately from the individual sensor error variances and temporal covariances, assuming that the individual observational errors of the three sensors in a MZ were not correlated with each other. The pooled error covariance matrix of the MZ-averaged soil moisture measurements indicated a significant autocorrelation of sensor observational errors of 0.518, as the systematic error standard deviation ($\sigma_{\overline{\alpha}} = 0.033$ $m^3$ $m^{-3}$) was similar to the random error standard deviation ($\sigma_{\overline{\epsilon}} = 0.032$ $m^3$ $m^{-3}$). To illustrate the impact of error covariance in probabilistic soil hydrological modeling, a case study was presented incorporating the pooled error covariance matrix in a Bayesian inverse modeling framework. These results demonstrate that the common assumption of uncorrelated random errors to determine parameter and model prediction uncertainty is not valid when measurements from sparse in situ soil moisture sensors are used to parameterize soil hydrological models. Further research is required to assess to what extent the error covariances found in this study can be transferred to other areas, and how they impact parameter estimation in soil hydrological modeling.

**Main abbreviations:** SWC, soil water content; MZ, measurement zone

**Keywords:** in situ measurements, observational errors, sensor measurement errors, soil moisture sensor, pooled errors, measurement zone, sparse measurements, pooled sensor calibration, error covariance matrix, Bayesian inverse modeling

**Graphical abstract**

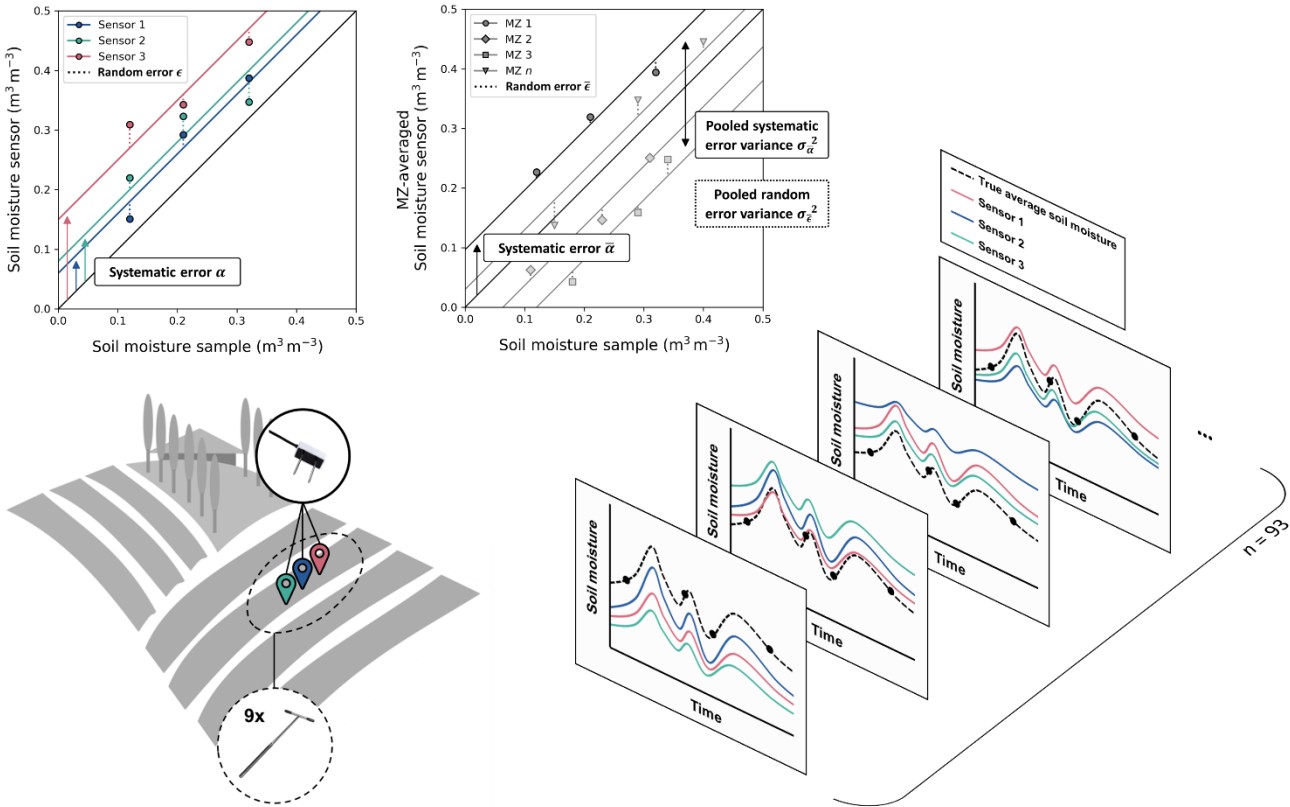

# 1 Introduction

Soil moisture measurements, such as measurements from in situ soil moisture sensors and sampling, are at the core of soil hydrological modeling, state and parameter estimation by assimilation, model validation, and decision making. However, these soil moisture measurements are subject to multiple sources of uncertainty, introducing systematic and random errors. Accurately quantifying these two types of errors is important to assess the uncertainty of estimated parameters and model predictions since the impact of random errors on this uncertainty vanishes with an increasing number of measurements whereas

that of the systematic errors does not. However, this error quantification presents a significant challenge.

Field-scale soil moisture patterns have a strong temporal stability which can be explained by spatial patterns in soil properties and topography (Brocca et al., 2010; Vachaud et al., 1985). As such, soil moisture observations at individual locations are characterized by time-stable statistical properties and some locations have the time-invariant property to represent the field mean (Vachaud et al., 1985), while other locations consistently deviate from this mean. Several studies have investigated an

optimal sampling or sensor network design to represent true soil moisture mean and variability in heterogeneous fields (Brocca et al., 2010; Chaney et al., 2015; Rossini et al., 2021; Wang et al., 2008), but such an optimal measurement design is not always feasible due to practical and budgetary constraints.

In addition to field-scale variability, microscale variability may also substantially impact soil moisture measurements (Hawley et al., 1983), especially point measurements with a small measurement volume. Microscale soil moisture variability may be

due to variations in soil particle and pore size, preferential flow (e.g., via biopores from burrowing animals), plant roots, microtopography, soil texture heterogeneity (e.g., clayey or sandy patches), uneven soil compaction, and localized irrigation practices (e.g., drip irrigation). As a result, soil moisture measurements may vary strongly depending on the location of the measurement (Schelle et al., 2013). When soil sampling is used to quantify soil moisture in a measurement zone (MZ) within a field, experimental errors can be minimized by collecting a composite sample from a sufficient number of random locations

within that MZ. While the measurement volume of a composite soil sample is large enough to average out microscale variability, a single sensor measurement is not and observational sensor measurement errors may depend on the local positioning of the sensors.

Observational sensor measurement errors include both inherent measurement errors (i.e., instrumental error arising from the measurement device, the measurement technique, environmental influence or signal processing) and representational errors

(i.e., spatial misrepresentation of the area or soil volume of interest). Quantifying such observational errors is trivial when measurements from sufficient locations are available. While experimental errors of subsequent soil moisture samplings over time are generally considered uncorrelated, such measurements are often temporally sparse. In contrast, using sensors allows for high temporal resolution, but typically only a few sensors are installed within a field often resulting in inadequate spatial coverage. This can lead to a biased mean sensor measurement compared to the true average soil moisture in the MZ, which

translates to autocorrelated sensor measurement deviations, i.e., observational errors that are correlated over time. This autocorrelation increases as the systematic error or bias becomes larger relative to the random error. Recently, Hendrickx et al. (2023) demonstrated that observational errors of soil moisture sensor measurements, i.e., the deviations between individual sensors and the true average soil moisture, are strongly correlated over time due to spatial variability and patterns in soil water retention properties.

Information on the spatiotemporal behavior of soil moisture measurements and their observational errors is especially important in the context of data assimilation and inverse modeling. Previous studies focused on spatial and temporal correlation of soil moisture measurements, as the required spatial density of the measurement network and the assimilation frequency depend on these properties, respectively (De Lannoy et al., 2006). Temporal correlation of soil water content (SWC) represents the persistence of SWC deviations from the long term temporal mean – a concept that is also referred to as 'soil moisture

memory' (Rahmati et al., 2024). This is related to the temporal dynamics of the meteorological forcings and to water flow in the soil, which depends on soil hydraulic properties. In this study, we are focusing on the temporal correlation of the

observational errors of soil moisture measurements, which we define as the deviations of soil moisture measurements from the mean soil moisture in the top 30 cm soil layer in a measurement zone (MZ) of about 80 m². This temporal correlation of errors is equal to the ratio of the error covariance between two points in time to the total error variance and is related to the temporal

stability of the spatial variability of soil moisture, rather than the temporal correlation of the SWC itself. We will refer to this temporal error correlation as error autocorrelation and will discuss potential implications of spatial correlation of the observational sensor errors on this error autocorrelation quantification.

The error covariance matrix quantifies both the magnitude of the observational errors (error variance on the diagonal) and how these errors are correlated across time (the off-diagonal elements are the error autocovariance), and is essential in data

assimilation as it helps to manage uncertainties and to correctly attribute weights to observational errors. Taking both observation and forecast bias into account in data assimilation, and estimating them in addition to or even simultaneous with model state variables results in improved estimation results, while neglecting error correlations can lead to significant errors in both the state and bias estimates, which in turn affects the overall model accuracy (Crow and Van Loon, 2006; Pauwels et al., 2013; Pauwels and De Lannoy, 2015).

The (log)likelihood function summarizes the errors between model simulations and corresponding observations, and incorporates uncertainties and error autocorrelations through the error covariance matrix. It plays a central role as an objective function in statistical modeling techniques, i.e., Bayes classifiers, support vector machines, Bayesian inverse modeling (e.g., Vrugt, 2016), and Bayesian data assimilation techniques such as an ensemble Kalman filter and particle filter (Wikle and Berliner, 2007). When using sensor measurements at fixed locations, autocorrelated observational errors need to be accounted

for. Residual errors are often both heteroscedastic and autocorrelated in hydrological modeling (Ammann et al., 2019; Evin et al., 2013; Samadi et al., 2018; Yang et al., 2007). However, most studies assume zero error covariance, and hence, often make incorrect assumptions on observational errors. For example, HYDRUS uses the Levenberg-Marquardt parameter estimation approach, which assumes a diagonal error covariance matrix (Šimůnek et al., 2012). Assuming zero error covariance is acceptable when an average of a large number of unbiased or calibrated sensors is used (e.g., Steenpass et al. (2010), who used

TDR sensors at 36 locations), but not if only a few sensors are available (e.g., Han et al. (2023)). Alternatively, error autocorrelation can be represented by autoregressive models, which have been assessed in several hydrological applications (Engeland and Gottschalk, 2002; Evin et al., 2013; Scharnagl et al., 2015).

When soil moisture is observed and modeled at sub-field or field-scale, limited methods exist to obtain a good estimate of the true mean soil moisture, its observational errors, and error autocorrelation. A MZ-specific error covariance matrix cannot be

derived accurately from a limited number of sensors in a field. Hendrickx et al. (2023) recently proposed a mechanistic error modeling approach to estimate soil moisture error (co)variabilities based on the spatial variability of the water retention curve. However, this method requires detailed soil data from repeated sampling of undisturbed soil cores, which is impractical. To the best of our knowledge, literature on this topic is scarce, hence further research is needed to address this gap.

We propose a pooled error modeling approach, which unifies observational errors that are identified in multiple fields with an

identical measurement setup but with only a limited number of sensors in each field. In this study, a pooled error covariance matrix is quantified based on a considerable dataset of sensor and soil sample data from 93 cropping cycles in agricultural fields in Flanders, Belgium (Sect. 2). This pooled error covariance matrix could then be applied in data assimilation or Bayesian inverse modeling across fields and soil types given the specific measurement setup as illustrated in Sect. 6, where parameters of an FAO-based soil water balance model are estimated using DREAM$_{(ZS)}$. First, the pooled sensor calibration is described in

Sect. 3. This calibration is applied to all sensor data prior to examining their observational errors. Then, the error model is described in Sect. 4, and is presented in two ways, i.e., using individual sensor measurements and using MZ averages. The quantification of the pooled errors is presented in Sect. 5.1-5.2, while the consequences of spatial sensor correlation are discussed in Sect. 5.3, and finally, the assumptions of the error model (i.e., data linearity, error normality, error stationarity,

spatial consistency and zero cross-correlation between soil samples and sensor measurements) are discussed in-depth in

Sect. 5.4.

**Table 1 List of symbols and their description**

| | |
|---|---|
| AR | Autocorrelation, i.e., temporal correlation, of observational errors |
| MZ | Measurement zone, i.e., a subplot within a field where measurements are taken |
| $S_{mV}$ | Raw sensor output (mV) |
| $s_{pooled}$ | Pooled standard deviation of individual soil moisture samples |
| SWC | Soil water content ($m^3 m^{-3}$) |
| $\alpha$ | Systematic error of an individual sensor |
| $\bar{\alpha}$ | Systematic error of an MZ-averaged sensor measurement |
| $\beta$ | Temporally variable process-related deviations between sensor measurements and the true soil moisture that are correlated between sensors |
| $\epsilon$ | Random error of an individual sensor |
| $\bar{\epsilon}$ | Random error of an MZ-averaged sensor measurement |
| $\epsilon_{nc}$ | Non-correlated random error of individual sensor measurements |
| $\theta_{g,samp}$ | Gravimetric SWC ($kg\ kg^{-1}$) of a soil gouge sample |
| $\theta_{sensor}$ | Calibrated sensor measurement ($m^3 m^{-3}$) representing the volumetric SWC in the 0-30 cm soil layer |
| $\theta_{sensor,nocal}$ | Volumetric SWC ($m^3 m^{-3}$) derived from sensor measurements calibrated with the manufacturer's calibration equation, but not calibrated against soil moisture measurements in the fields |
| $\theta_{v,samp}$ | Volumetric SWC ($m^3 m^{-3}$) of a soil gouge sample |
| $\rho_b$ | Dry bulk density ($kg\ m^{-3}$) |
| $\rho_\alpha$ | Temporally stable spatial sensor correlation, i.e., correlation between systematic errors of individual sensors |
| $\rho_\epsilon$ | Temporally variable spatial sensor correlation, i.e., correlation between the 'random' errors of individual sensors |
| $\sigma_{samp}^2$ | Pooled error variance of composite soil moisture samples |
| $\sigma_{tot}^2$ | Pooled total error variance of an individual sensor |
| $\sigma_{\overline{tot}}^2$ | Pooled total error variance of the MZ-averaged sensor measurements |
| $\sigma_\alpha^2$ | Pooled systematic error variance of an individual sensor |
| $\sigma_{\bar{\alpha}}^2$ | Pooled systematic error variance of the MZ-averaged sensor measurements, i.e., pooled error covariance |
| $\sigma_\epsilon^2$ | Pooled random error variance of an individual sensor |
| $\sigma_{\bar{\epsilon}}^2$ | Pooled random error variance of the MZ-averaged sensor measurements |

## 2    Study sites and data

Each year during three growing seasons (2021−2023), about 30 agricultural fields for vegetable production were equipped with a sensor module (Fig. 1). In every field, soil moisture samples were taken on a regular basis. All fields were located in

Flanders, the northern half of Belgium, had an area of 1 to 5 ha, were irrigated using various irrigation methods, and included soil textures ranging from sand to silt loam. While most of the fields were for commercial production purposes, experimental fields at three research centers were included as well.

Dielectric capacitance soil moisture sensors (TEROS 10, Meter Group, Inc., USA) were used to measure daily volumetric SWCs in the fields. The sensors use an electromagnetic field to measure the dielectric permittivity of the surrounding medium

within a measurement volume of 430 mL, approximately corresponding to a cylinder with a diameter of 7.1 cm and a height of 10.9 cm. A sensor module (Agrisense Pro, Io-Things, Belgium) consisted of three TEROS 10 sensors connected to a datalogger equipped with a communication module (Sigfox). The communication module enabled the acquisition and transmission of sensor data to an online server, ensuring real-time online data access. The sensors were installed horizontally at 15 cm depth in a straight line with 2 m distance between each consecutive sensor within the MZ specified by the farmer.

Since the sensors were connected to a datalogger of Io-Things, the calibration equation for third-party loggers (Eq. (1)) was

applied to convert the raw sensor output in mV to volumetric SWC (m³ m⁻³), rather than the manufacturer's calibration equation designed for METER loggers (TEROS 10, 2024).

$$\theta_{\text{sensor,nocal}} = -2.154 + 3.898 \times 10^{-3} \times S_{\text{mV}} - 2.278 \times 10^{-6} \times S_{\text{mV}}^2 + 4.824 \times 10^{-10} \times S_{\text{mV}}^3 , \tag{1}$$

where $S_{\text{mV}}$ is the raw sensor output (mV), and $\theta_{\text{sensor,nocal}}$ is the volumetric SWC (m³ m⁻³) derived from sensor measurements that were not calibrated against soil moisture measurements in the fields. A list of symbols used in this paper is provided in Table 1.

At the beginning of the growing season, undisturbed Kopecky ring samples (V: 100 cm³, h: 51 mm) were taken from the 10-15 cm depth to determine bulk density. Three ring samples were collected per MZ. Soil moisture samples (from 2 to 30 cm depth) were taken regularly (every two to four weeks) with a gouge auger at all sites during the growing period, and soil moisture was quantified using the gravimetric method. The volumetric SWC was then calculated based on the gravimetric SWC and bulk density (Eq. (2)).

$$\theta_{\text{v,samp}} = \theta_{\text{g,samp}} \frac{\rho_{\text{b}}}{\rho_{\text{w}}} , \tag{2}$$

where $\theta_{\text{v,samp}}$ is the volumetric SWC (m³ m⁻³) and $\theta_{\text{g,samp}}$ is the gravimetric SWC (kg kg⁻¹) of the gouge samples, $\rho_{\text{b}}$ is the dry bulk density (kg m⁻³) and $\rho_{\text{w}}$ is the mass density of water (kg m⁻³). At all sites, multiple soil moisture samples (nine in commercial fields, six in experimental fields) were collected within a radius of 5 m around the sensors (Fig. 1). These samples were generally combined into a composite sample, while at some of the sites, each sample was analyzed individually to obtain an accurate estimate of the soil moisture sample errors (Sect. 4.1).

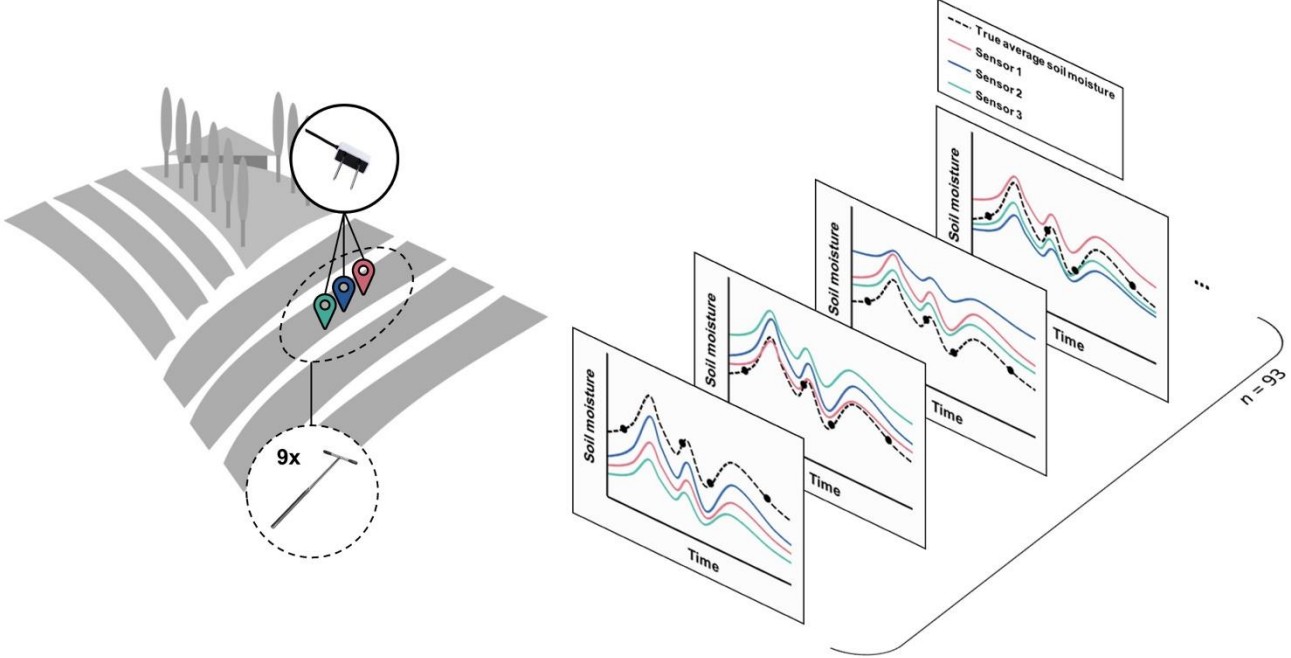

**Fig. 1 Illustration of the measurement setup in agricultural fields: The true SWC of the MZ is represented by a composite soil moisture sample of 9 individual gouge auger samples, while three fixed soil moisture sensors measured SWC at 15 cm depth. During three growing seasons (2021−2023), measurement data were collected of 93 cropping cycles.**

During data preprocessing, only daily sensor measurements where data from all three sensors were available were retained for error quantification. For fields where two cropping cycles were monitored within the same year, the data were split into two separate cropping cycles as the sensors were removed and reinstalled. Then, cropping cycles that had fewer than two soil moisture sampling events conducted in parallel with the sensor data were excluded from the analysis to ensure the reliability and accuracy of the error quantification. These preprocessing steps resulted in 93 cropping cycles that were retained for analysis (Fig. 1).

## 3    Pooled sensor calibration

In addition to the manufacturer's calibration equation, a pooled linear recalibration was established to relate the point measurements at 15 cm depth by the TEROS 10 sensors with soil moisture samples measuring the whole upper 30 cm layer, so as to obtain sensor measurement data that are representative for this upper soil layer. The composite soil moisture samples of the 30 cm layer are plotted against their corresponding mean sensor measurements at 15 cm depth from the same MZ and time points for all study sites in 2021, 2022 and 2023 (Fig. 2). A bias (ME) of -0.043 $m^3$ $m^{-3}$ and an RMSE of 0.058 $m^3$ $m^{-3}$ was observed, indicating a significant underestimation of SWC in the upper 30 cm layer of the MZ by the three sensors at 15 cm depth.

The calibration curve was fitted using an orthogonal Deming regression, as both the sensor measurements and soil moisture samples are subject to measurement uncertainty (Deming, 1938; Ludbrook, 2010). In this regression method, the squares of the perpendicular distances of the calibration points from the regression line are minimized. The prerequisites for this regression method include identical scales of the $x$ and $y$ variables, similar error variances of the $x$ and $y$ variables, and a correlation coefficient close to 1, all of which were satisfied for our measurement dataset. The data covered a wide range of SWCs and were strongly correlated with a Pearson correlation of 0.83. The resulting calibration curve (Eq. (3)) had an $R^2$ of 0.67 and an RMSE of 0.043 $m^3$ $m^{-3}$ (Fig. 2). The (perpendicular) residual plot shows randomly scattered residuals and a constant variance, suggesting homoscedasticity (Appendix A: Fig. A1).

$$\theta_{\text{sensor}} = -0.006 + 1.26 \times \theta_{\text{sensor,nocal}} \, , \tag{3}$$

where $\theta_{\text{sensor}}$ ($m^3$ $m^{-3}$) is the calibrated sensor measurement representing the volumetric SWC in the 0-30 cm soil layer, and $\theta_{\text{sensor,nocal}}$ ($m^3$ $m^{-3}$) is the volumetric SWC that is measured by the non-calibrated sensors at 15 cm depth by applying Eq. (1). The pooled sensor calibration was applied to all sensor data before examining the observational errors.

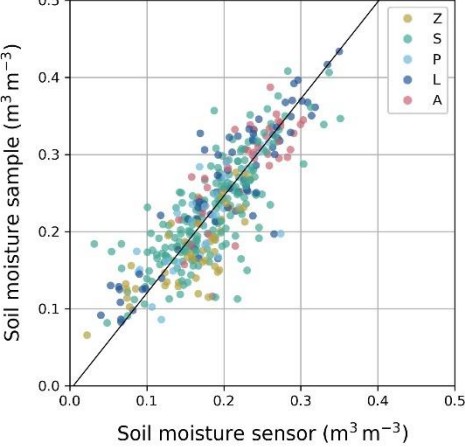

Fig. 2 Mean soil moisture samples, $\theta_{\text{v,samp}}$ ($m^3$ $m^{-3}$), in function of mean non-calibrated soil moisture sensor measurements, $\theta_{\text{sensor,nocal}}$ ($m^3$ $m^{-3}$), with the pooled sensor calibration curve to obtain sensor measurement data that are representative for the top 30 cm soil layer, as represented by soil moisture samples. The observations are color-coded based on Belgian soil texture class (Z: Sand, S: Loamy sand, P: Light sandy loam, L: Heavy sandy loam, A: Silt loam).

A lab-based calibration of the TEROS 10 sensor can be found in Supplementary Materials (S1), but was not applied in this study. However, the similarity between the lab-based sensor calibration ($\theta = -0.013 + 1.16 \times \theta_{\text{sensor,nocal}}$) and the field-based pooled sensor calibration (Eq. (3)) suggests that this pooled calibration has a broader applicability, e.g., on different fields and in different contexts, and that the calibration mainly corrects for soil moisture measurement inaccuracy rather than the discrepancy between representative measurement volumes of the soil sample (0-30 cm depth) and the sensor (soil volume of 430 mL at 15 cm depth). Mane et al. (2024) state that pooled ('generalized') sensor calibrations, i.e., using measurements from multiple sites across a large region, are a viable alternative to field- or soil-specific sensor calibrations, but note that the accuracy is lower.

## 4 Pooled error model approach

### 4.1 Soil moisture sample variance

The pooled variance of composite soil moisture samples ($\sigma_{\text{samp}}^2$) can be determined based on sampling events during which multiple soil moisture samples, i.e., multiple punctures with the gouge auger from the same MZ, are analyzed individually. First, the sample standard deviation $s$ (for individual samples) can be quantified for each multi-sampling event in each MZ. Then, the pooled standard deviation can be computed to obtain a weighted average of all standard deviations by using Eq. (4), to represent the standard deviation of individual soil moisture samples, assuming that the standard deviation of individual soil

samples is spatially and temporally constant. However, it is known that the true soil moisture variability is likely dependent on the spatial variability of soil properties and SWC itself (e.g., Hendrickx et al., 2023), which is not accounted for here.

$$s_{\text{pooled}} = \sqrt{\frac{(n_1-1)s_1^2 + (n_2-1)s_2^2 + \dots + (n_p-1)s_p^2}{n_1 + n_2 + \dots + n_p - p}} , \tag{4}$$

where $s_{\text{pooled}}$ is the pooled standard deviation, $p$ is the number of sampling events, and $s_i$ and $n_i$ are the sample standard deviation and sample size of the $i^{\text{th}}$ sampling event, respectively. Finally, the standard error of a composite sample consisting of $n$ individual samples can be computed by dividing the pooled standard deviation by the square root of $n$, resulting in the

pooled variance of composite soil moisture samples as given by Eq. (5).

$$\sigma_{\text{samp}}^2 = \frac{s_{\text{pooled}}^2}{n} , \tag{5}$$

where $n$ is the number of individual samples in the composite soil sample.

Since soil moisture sample errors are assumed to be mainly attributed to spatial variability of soil moisture, assuming that sample measurement errors (such as handling errors or incomplete oven drying) are minimal, a multi-sample analysis is recommended to obtain a more accurate estimate of the soil moisture sample errors for a specific field or MZ. Additionally,

the pooled error model approach assumes zero cross-correlation between the errors of the soil moisture samples and the observational errors of sensor measurements.

### 4.2 Sensor error model

#### 4.2.1 Individual sensor measurements

When repeated measurements from a sufficiently large number of sensors are available in a MZ, the error covariance can be

quantified based on the soil moisture measurements directly. However, Western and Blöschl (1999) stated that bias is introduced in spatial statistical properties of soil moisture such as covariance and correlation length as the spatial coverage ('extent') of soil moisture measurements decreases. Hence, when a small set of sensors has limited spatial coverage, the variability in the MZ cannot be accurately described by these sensors, which results in an underestimation of the (co)variability, and the mean sensor measurement may be biased compared to the true mean SWC due to local differences. In this case, a

direct calculation may not provide accurate estimates of true SWC error variability and autocorrelation. The limited number of sensors also directly translates to wide confidence intervals on the covariance estimate due to the limited degrees of freedom, e.g., d.f. = 2. Alternatively, a pooled error model approach is proposed that uses higher degrees of freedom by combining information from multiple measurement sites, and is based on a commonly used error model formulation with an additive systematic error term (bias) and a random error term (Eq. (6)).

$$\theta_{\text{sensor},i,k} = \bar{\theta}_i + \alpha_k + \epsilon_{i,k} , \tag{6}$$

where $\theta_{\text{sensor},i,k}$ is the calibrated SWC measured at time $i$ by sensor $k$ (using Eq. (3)), $\bar{\theta}_i$ is the 'true' mean SWC derived from the soil sample measurements at time $i$, $\alpha_k \sim \mathcal{N}(0, \sigma_\alpha^2)$ is a systematic error or bias specific to sensor $k$, and $\epsilon_{i,k} \sim \mathcal{N}(0, \sigma_\epsilon^2)$ is a random error (Fig. 3a). It is important to note that, in order to compute a pooled (co)variance, the model assumes that the (co)variances are equal over time and across different MZs, reflecting temporal and spatial consistency. The systematic error

is the time-invariant component of the deviation of the sensor measurement $\theta_{\text{sensor},i,k}$ from the true SWC from the sampling, while the random error is the time-variant component. No multiplicative systematic error is considered here, as this has already been addressed by applying the pooled sensor calibration (Eq. (3)). When there is only a small number and a limited range of composite soil moisture samples available over time in each MZ, a sensor- or MZ-specific slope cannot be derived.

The systematic error ($\alpha$) of a sensor in a MZ corresponds with its sensor-specific intercept of the relation between the sensor measurement and the 'true' SWC derived from the soil samples ($\hat{\theta}_{i,k} = \bar{\theta}_i + \alpha_k$), as illustrated in Fig. 3a. The pooled systematic error variance ($\sigma_\alpha^2$) can be calculated from the sensor-specific intercepts of all sensors that are installed in all fields (Eq. (7)).

$$\sigma_\alpha^2 = \text{var}(\alpha) = \frac{1}{S-1}\sum_{k=1}^{S} \alpha_k^2 \, , \tag{7}$$

where $S$ is the number of sensor-specific intercepts.

Then, the pooled random error variance ($\sigma_\epsilon^2$) is defined as the variance of the sensor measurement deviations, $\epsilon$, with respect to their sensor-specific curve ($\hat{\theta}_{i,k} = \bar{\theta}_i + \alpha_k$), as illustrated in Fig. 3a, using Eq. (8).

$$\sigma_\epsilon^2 = \frac{\sum_{k=1}^{S}\sum_{i=1}^{N_k}\left(\theta_{\text{sensor},i,k}-\hat{\theta}_{i,k}\right)^2}{\sum_{k=1}^{S} N_k - S} \, , \tag{8}$$

where $S$ is the number of sensor-specific intercepts and $N_k$ is the number of data points measured by sensor $k$, while $\theta_{\text{sensor},i,k}$ is the calibrated SWC measured by sensor $k$ at time $i$, and $\hat{\theta}_{i,k}$ is the expected SWC measured by sensor $k$ at time $i$ based on their sensor-specific curve.

Finally, the total error variance, $\sigma_{\text{tot}}^2$, of soil moisture measurements by an individual sensor is defined as the sum of the pooled systematic and random error (Eq. (9)).

$$\sigma_{\text{tot}}^2 = \sigma_\alpha^2 + \sigma_\epsilon^2 \, . \tag{9}$$

According to the error model in Eq. (6), the autocovariance of the observational errors is equal to the systematic error variance, $\sigma_\alpha^2$, as derived in Appendix B. The autocorrelation (AR) between observational errors at two moments is quantified as the ratio of the autocovariance and the total error variance (Eq. (10)), assuming that the standard deviation of the errors are constant over time, and that error autocorrelation is equal for any pair of measurements.

$$\text{AR} = \frac{\sigma_\alpha^2}{\sigma_{\text{tot}}^2} \, . \tag{10}$$

In this error model, the pooled error variance, covariance and autocorrelation are assumed to be constant in time and equal for all sensors in all fields. As they are pooled over different fields with different soil types, they are expected to be applicable to all fields in the area of Flanders without significant topography or heavy soils and with a specific measurement setup. Hence, we assume that there are no variations in error variance, covariance, and autocorrelation between different fields, e.g., due to varying soil properties and soil heterogeneity between different fields, nor variations due to varying soil moisture states. This is a strong assumption that will be further discussed in Sect. 5.4 Using a direct calculation method, a sufficient number of sensors ($n$) is required to accurately represent field and state-dependent error variances and covariances. The pooled approach, on the other hand, offers the advantage of not being constrained by the number of sensors in a single field, making it suitable for scenarios where sensor deployment is limited.

### 4.2.2 Averaged sensor measurements

An analogous error model can be formulated for the average of the soil moisture sensor measurements in a MZ with multiple sensors (Eq. (11)).

$$\bar{\theta}_{\text{sensor},i,f} = \bar{\theta}_i + \bar{\alpha}_f + \bar{\epsilon}_{i,f} \, , \tag{11}$$

where $\bar{\theta}_{\text{sensor},i,f}$ is the average SWC measured at time $i$ by the sensors in field $f$, $\bar{\theta}_i$ is the 'true' mean SWC in the MZ derived from the soil sample measurements, $\bar{\alpha}_f \sim \mathcal{N}(0,\sigma_{\bar{\alpha}}^2)$ is a systematic error and $\bar{\epsilon}_{i,f} \sim \mathcal{N}(0,\sigma_{\bar{\epsilon}}^2)$ is a random error (Fig. 3b).

Now, the systematic error ($\bar\alpha$) of a MZ corresponds with its MZ-specific intercept, which is the average of the intercepts of the individual sensors in that MZ. The variance of all MZ-specific intercepts corresponds to the pooled systematic error variance, or error covariance ($\sigma_{\bar\alpha}^2$, Eq. (12)). This pooled systematic error variance is illustrated in Fig. 3b.

$$\sigma_{\bar\alpha}^2 = \text{var}(\bar\alpha) = \frac{1}{F-1}\sum_{f=1}^{F}\bar\alpha_f^2 , \tag{12}$$

where $F$ is the number of fields.

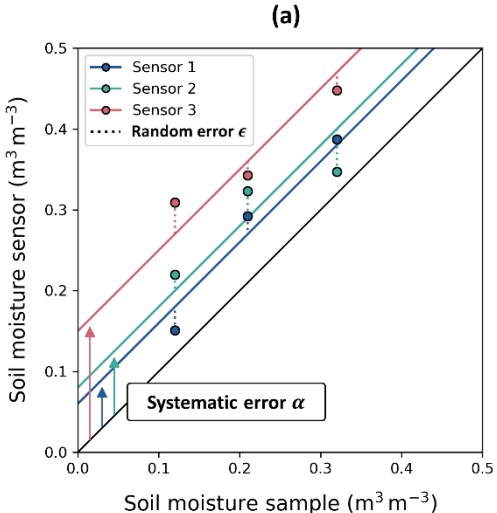

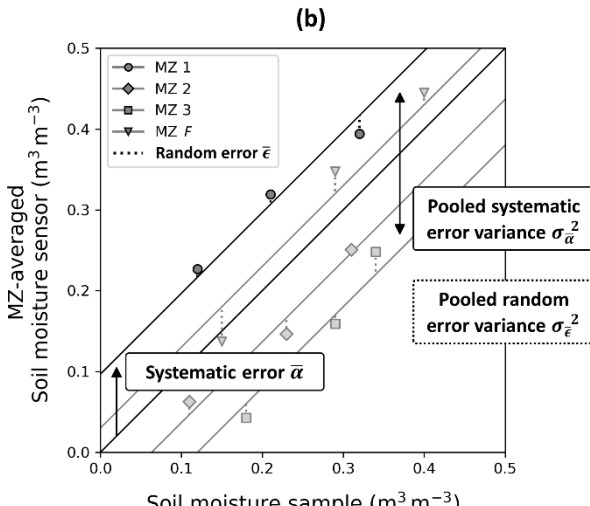

Fig. 3 Observational errors of (a) individual sensor measurements, and (b) MZ-averaged sensor measurements with their error variances. The three sensors in (a) correspond to MZ 1 in (b).

The pooled random error variance ($\sigma_{\bar\epsilon}^2$) is defined as the variance of the deviations of the average sensor measurements, $\bar\epsilon$, with respect to their MZ-specific curve ($\hat\theta_{i,f} = \bar\theta_i + \bar\alpha_f$) using Eq. (13).

$$\sigma_{\bar\epsilon}^2 = \frac{\sum_{f=1}^{F}\sum_{i=1}^{N_f}\left(\bar\theta_{\text{sensor},i,f} - \hat\theta_{i,f}\right)^2}{\sum_{f=1}^{F}N_f - F} , \tag{13}$$

where $F$ is the number of fields, and $N_f$ is the number of data points in field $f$.

The total error variance of the averaged soil moisture measurement, $\sigma_{\overline{\text{tot}}}^2$, is defined as the sum of the pooled systematic and random error variances (Eq. (14)):

$$\sigma_{\overline{\text{tot}}}^2 = \sigma_{\bar\alpha}^2 + \sigma_{\bar\epsilon}^2 . \tag{14}$$

When all sensors in a MZ are (spatially) independent from each other, the variances of the systematic and random errors of the MZ-averaged soil moisture measurements are related to the respective error variances obtained with Eqs. (7)-(8)(9) as given by Eqs. (15)-(17), respectively (as derived in Appendix C, Eq. (C1)-(C3)).

$$\sigma_{\bar{\alpha}}^2 = \frac{\sigma_\alpha^2}{n},$$ (15)

$$\sigma_{\bar{\epsilon}}^2 = \frac{\sigma_\epsilon^2}{n},$$ (16)

$$\sigma_{\overline{\text{tot}}}^2 = \frac{\sigma_{\text{tot}}^2}{n},$$ (17)

where $n$ is the number of sensors in a MZ. The autocorrelation (AR) of the errors of the mean of all sensors in a MZ is given by Eq. (18). As a result, for independent sensors, the autocorrelation (AR) of the errors of the MZ-averaged soil moisture measurements is equal to that of the individual sensor measurements (Eq. (10)).

$$AR = \frac{\sigma_{\bar{\alpha}}^2}{\sigma_{\overline{\text{tot}}}^2}.$$ (18)

### 4.3 Spatial sensor correlation within a MZ

Spatial dependency is inherently present within a MZ as locations that are closer together tend to have more similar soil and plant properties. Therefore, it is best to distribute soil sample and sensor locations well spread across the field or MZ. However, it is not practical to use a set of sensors that are connected to an IoT datalogger via long sensor cables, as they can complicate sensor installation and hinder field operations. In case a small set of sensors with short sensor cables is installed in close proximity within a MZ and the observational errors of the sensors, i.e., the deviations of the point measurements compared to the true average SWC in the top 30 cm soil layer of a MZ, are spatially correlated with each other, we underestimate the observational errors when using Eq. (15)-(16) to infer the systematic and random error variances of MZ-averaged moisture measurements from systematic and random error variances of the individual sensor measurements.

Spatial sensor correlation can be divided in a temporally stable spatial SWC pattern and a spatial correlation of temporal deviations from the stable SWC pattern. Temporally stable spatial correlation between measurement points that are close to each other within a MZ manifests itself as a systematic deviation of the sensor-specific intercepts ($\alpha$) within a MZ (illustrated in Fig. 4a) so that $\sigma_{\bar{\alpha}}^2 > \frac{\sigma_\alpha^2}{n}$, where $n$ is the number of sensors in a MZ. The correlation between the $n$ sensor-specific intercepts in a MZ, $\rho_\alpha$, can be quantified using Eq. (19) (as derived in Appendix C). Due to the stable spatial SWC pattern, the systematic deviation of a sensor set of three sensors will depend on the specific location of the sensors within the MZ so that two different sensor sets in the same MZ might have a different systematic deviation from the MZ average SWC (Fig. 4a).

$$\rho_\alpha = \frac{n\sigma_{\bar{\alpha}}^2 - \sigma_\alpha^2}{(n-1)\sigma_\alpha^2}.$$ (19)

The degree of spatial correlation can be assessed in three ways. The first method involves constructing a semivariogram by quantifying spatial soil moisture variability for different distances. Measured soil moisture variability is expected to increase with distance until soil moisture semivariances stabilize, at which point the measurements can be considered independent and the correlation length, i.e., the range of spatial dependence, can be roughly estimated. The second method compares the variability of the systematic errors $\alpha$ obtained per field ($\sigma_{\bar{\alpha}}^2$), i.e., from the average sensor measurements, and the variability of the systematic errors $\alpha$ obtained from the individual sensors ($\sigma_\alpha^2$), both qualitative and numerical using Eq. (19). Spatial independence of $n$ sensors within a field implies that the variance of sensor-specific intercepts ($\sigma_\alpha^2$) equals $n$ times the variance of MZ-specific intercepts ($\sigma_{\bar{\alpha}}^2$) (Eq. (15)). Deviations from this condition indicate temporally stable spatial dependence among sensors. The third method analyzes the spatial correlation between the sensor-specific intercepts per MZ.

Analogously, a spatial correlation of random errors of individual sensors, $\epsilon$, corresponds to a spatial correlation of temporal variations in SWC (illustrated in Fig. 4b). These temporally varying deviations are related to soil hydrological processes that change soil moisture. Spatial covariance of hydrological processes and of soil properties (soil texture, soil structure, organic

matter content, bulk density, and hydraulic conductivity) that define how SWC changes in response to a process result in a spatial covariance of temporal variations in SWC. One can expect that all sensors at 15 cm depth would measure a lower SWC compared to the 'true' SWC of the 0-30 cm soil layer just after a rainfall or irrigation event because precipitation in the top layer has not (yet) been detected by the sensors. Similarly, all sensors would measure a higher SWC compared to the 'true' SWC of the 0-30 cm soil layer in periods with high evaporation from the top soil layer. This results in a random error that varies over time, but part of this variation will be similar for the different sensors in the MZ, i.e., part of the temporal variation of $\epsilon$ in Eq. (8) is 'shared' or correlated among sensors. This shared temporal variation of differences between sensor measurements and the true mean could be represented in the error model by introducing Eq. (6) with a temporally varying term $\beta_i$ that is equal for all sensors at time $i$, while $\epsilon_{nc}$ is the remaining non-correlated part of the random error (Eq. (20)).

$$\theta_{sensor,i,k} = \bar{\theta}_i + \alpha_k + \beta_i + \epsilon_{nc,i,k} \ . \tag{20}$$

In contrast to $\alpha$, which could be interpreted as a temporally fixed deviation related to the spatial variation, the temporally varying $\beta$ represents process-related deviations between sensor measurements and the true SWC that are correlated between sensors. The temporally variable spatial sensor correlation would manifest itself as a systematic deviation of all sensors in a field at a certain time step (Fig. 4b). When sensor measurements in a MZ are averaged, the deviation of the spatial average from the true SWC mean that is corrected for the average of the systematic deviations of the sensors, $\bar{\epsilon}$, contains both $\beta$ and a non-correlated random error $\epsilon_{nc}$:

$$\bar{\epsilon}_{i,f} = \beta_i + \frac{1}{n}\sum_{k=1}^{n} \epsilon_{nc,i,k} \ , \tag{21}$$

so that:

$$\sigma_{\bar{\epsilon}}{}^2 = \sigma_{\beta}^2 + \frac{\sigma_{\bar{\epsilon},nc}^2}{n} > \frac{\sigma_{\beta}^2}{n} + \frac{\sigma_{\bar{\epsilon},nc}^2}{n} = \frac{\sigma_{\epsilon}^2}{n} \ . \tag{22}$$

In a sensor setup with $n$ perfectly correlated sensors, the total random error variance of the MZ averages ($\sigma_{\bar{\epsilon}}{}^2$) will be equal to the total random error of the individual sensors ($\sigma_{\epsilon}^2$) (illustrated in Fig. 4b). The correlation between the 'random' errors of the individual sensors can be quantified using Eq. (23) as derived in Appendix C. This correlation is equal to the ratio of the sensor-correlated 'random' error (co)variance ($\sigma_{\beta}^2$) to the total 'random' error variance ($\sigma_{\epsilon}^2$).

$$\rho_{\epsilon} = \frac{n\sigma_{\bar{\epsilon}}{}^2 - \sigma_{\epsilon}^2}{(n-1)\sigma_{\epsilon}^2} = \frac{\sigma_{\beta}^2}{\sigma_{\epsilon}^2} \ . \tag{23}$$

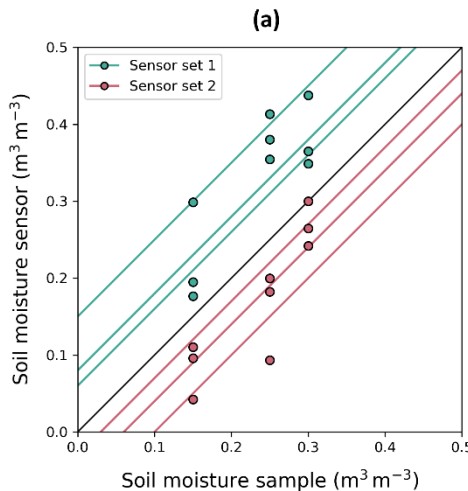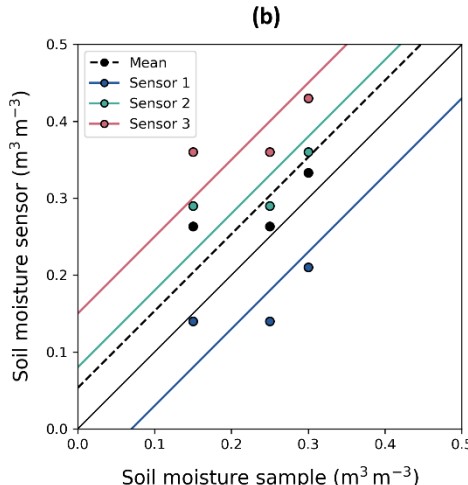

Fig. 4 (a) Illustration of spatial correlation of sensors in a MZ: Sensors that are close together have a similar deviation from the average of the MZ. Two sets of three sensors in the same MZ might have different systematic deviations depending on their location. (b) Illustration of perfect (temporally variable) process-related sensor correlation: The three sensors in a single MZ show equal deviations from their sensor-specific curve for a certain soil moisture sampling event.

## 4.4 Pooled error covariance matrix of averaged soil moisture measurements in a MZ

When all pooled errors are quantified, the pooled error covariance matrix can be built. The error covariance matrix for field $f$ has a size $M \times M$, with $M$ being the sum of the number of (daily) mean sensor measurements over time ($N_f$) and the number of soil moisture sample events ($p$) in field $f$. The error covariance matrix (Fig. 5) contains a $N_f \times N_f$ matrix with the pooled total error variance $\sigma_{\overline{tot}}^2$ (Eq. (14)) on the diagonal and the pooled error covariance $\sigma_{\overline{\alpha}}^2$ (Eq. (12)) off-diagonal. The additional $p$ rows and columns represent the uncorrelated composite soil moisture sample variabilities, with the pooled sample variance $\sigma_{samp}^2$ (Eq. (5)) on the diagonal and off-diagonal zeros. The pooled error covariance matrix is by definition invertible and well-conditioned as long as a significant random error is present, i.e., the observational errors are not perfectly autocorrelated.

| | 1 | 2 | 3 | ... | $N_f$ | 1 | ... | $p$ |
|---|---|---|---|---|---|---|---|---|
| **1** | $\sigma_{\overline{tot}}^2$ | $\sigma_{\overline{\alpha}}^2$ | $\sigma_{\overline{\alpha}}^2$ | ... | $\sigma_{\overline{\alpha}}^2$ | 0 | 0 | 0 |
| **2** | $\sigma_{\overline{\alpha}}^2$ | $\sigma_{\overline{tot}}^2$ | $\sigma_{\overline{\alpha}}^2$ | ... | $\sigma_{\overline{\alpha}}^2$ | 0 | 0 | 0 |
| **3** | $\sigma_{\overline{\alpha}}^2$ | $\sigma_{\overline{\alpha}}^2$ | $\sigma_{\overline{tot}}^2$ | ... | $\sigma_{\overline{\alpha}}^2$ | 0 | 0 | 0 |
| **...** | ... | ... | ... | ... | $\sigma_{\overline{\alpha}}^2$ | 0 | 0 | 0 |
| **$N_f$** | $\sigma_{\overline{\alpha}}^2$ | $\sigma_{\overline{\alpha}}^2$ | $\sigma_{\overline{\alpha}}^2$ | $\sigma_{\overline{\alpha}}^2$ | $\sigma_{\overline{tot}}^2$ | 0 | 0 | 0 |
| **1** | 0 | 0 | 0 | 0 | 0 | $\sigma_{samp}^2$ | 0 | 0 |
| **...** | 0 | 0 | 0 | 0 | 0 | 0 | $\sigma_{samp}^2$ | 0 |
| **$p$** | 0 | 0 | 0 | 0 | 0 | 0 | 0 | $\sigma_{samp}^2$ |

Fig. 5 Error covariance matrix for $N_f$ days of sensor measurements and $p$ soil moisture sampling events in field $f$.

## 5 Results and Discussion

### 5.1 Uncorrelated soil moisture samples

The pooled standard deviation (Eq. (4)) for individual soil moisture samples was 0.0114 m³ m⁻³. For nine individual soil samples in a sampling event, the standard error of the mean was 0.0038 m³ m⁻³ ($\sigma_{samp}^2 = 0.0000144$). This standard error was small enough to consider these soil moisture samples as reliable reference measurements.

The standard deviation of individual soil moisture samples was smaller than soil moisture sampling variabilities found in literature (e.g., Brocca et al., 2010; Famiglietti et al., 2008). However, it is possible that different sampling depths and methods, but also differences in heterogeneity, result in different sampling variabilities. In a particularly heterogeneous field or MZ, a multi-sample analysis is recommended to obtain a more accurate estimate of the soil moisture sample variance specifically for that MZ.

### 5.2 Quantifying observational errors of sensor measurements

#### 5.2.1 Systematic errors

In a field equipped with sensor sets in multiple MZs within the field, each sensor set was characterized by a different systematic deviation, $\alpha$, from their MZ-specific mean SWC measured by the soil moisture samples, of which two MZs are shown in Fig. 6. Moreover, all three sensors in a given MZ showed similar deviations from the composite soil moisture sample in that MZ, e.g., the sensors in MZ 2 all measured a consistently higher SWC compared to the soil moisture samples (Fig. 6b). These

similar deviations demonstrate that the sensors within a MZ are not independent, but rather have observational errors that are spatially correlated. Fig. 6a also shows that the autocorrelation of the observational errors remains persistent over time, which is not in line with a classical autoregressive model, where a decay with increasing time lag is typically expected. Moreover, if these error autocorrelations were to be time-variable, autocorrelation would be a function of SWCs or (soil) hydrological events rather than time lag or time itself (Hendrickx et al., 2023).

Furthermore, the sensor data may underestimate the true average SWC in a certain MZ (Fig. 6, MZ 1), while in another MZ, the sensor data may overestimate the true average SWC in the MZ (Fig. 6, MZ 2). This suggests that if the sensor set were installed at a different position within the same MZ, the systematic deviation, $\alpha$, from its MZ-specific mean SWC would be different (as is also illustrated by sensor removal and reinstallation in Appendix A: Fig. A2). Hence, when more sensors would be installed within a MZ, expanding spatial coverage and reducing spatial correlation of observational errors (Eq. (19) and

(23)), the systematic error of the MZ-average is expected to decrease, likely causing a reduced error autocorrelation (AR) of the average sensor measurement.

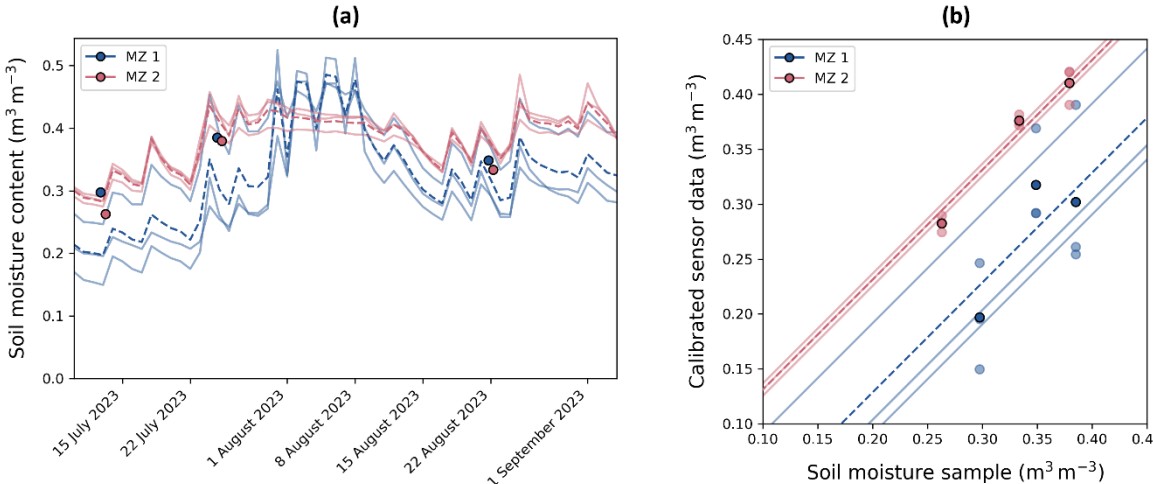

**Fig. 6 (a) SWC measured in two MZs within a field with three sensors per MZ. The sensor data were calibrated with the pooled sensor calibration (Eq. (3)), and are plotted along with the MZ-averaged SWC (- - -). (b) Mean SWC (0-30cm) measured with a**
**composite soil sample are plotted against the mean sensor measurements at each location, and their MZ-specific regression line with a slope equal to 1 is shown. The MZ-averaged SWCs and curves are also shown (- - -).**

The pooled systematic error variance was quantified for the individual sensors as well as the averaged sensor measurements based on all cropping cycles (Table 2). The individual sensor measurements resulted in 279 sensor-specific intercepts, all based on more than one sampling event, as shown in Fig. 7, which illustrates that measurements by a single sensor may differ

consistently over time from the true soil moisture in the top 30 cm soil layer. The standard deviation of these intercepts was 0.037 m³ m⁻³, which corresponded to an error covariance of an individual sensor of $\sigma_\alpha^2 = 0.001380$. Under the assumption of sensor independence and for three sensors in a MZ, the error covariance of the average measured soil moisture would be $\sigma_{\overline{\alpha}}^2 = 0.000460$ (Eq. (15)).

Next, the 93 MZ-specific intercepts, all based on the averages of three soil moisture sensor measurements and more than one
sampling event, were estimated. The standard deviation of these intercepts was 0.033 m³ m⁻³, which corresponded to an error covariance of $\sigma_{\overline{\alpha}}^2 = 0.001070$, and was considerably larger than the estimate based on the assumption of non-correlated systematic errors.

When analyzing double cropping cycles on a certain field within one year, we see how the mean bias (intercept) shifts after the sensors are removed and reinstalled (Appendix A: Fig. A2). This demonstrates the impact of sensor repositioning on
measurement accuracy, highlighting the systematic changes that can occur due to sensor position adjustments.

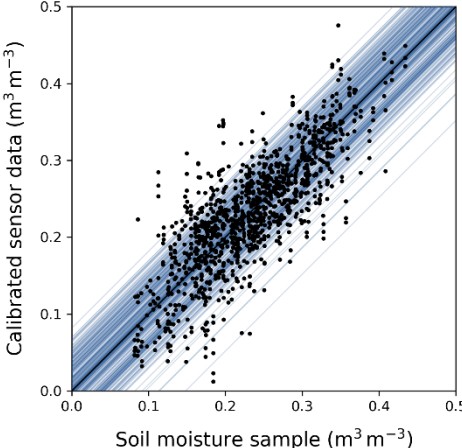

**Fig. 7 Intercepts based on individual sensor measurements resulting in 279 sensor-specific curves.**

Systematic errors between the mean soil moisture measurement obtained from few sensors with limited spatial coverage and the true mean soil moisture of the MZ may originate from time-persistent spatial differences in soil moisture. Such time-persistent spatial differences may be due to variability in soil properties, water retention, vegetation cover and root distribution (Hendrickx et al., 2023; Schlüter et al., 2013), as well as groundwater depth, topography, and non-equilibrium (Vogel et al., 2010; Schlüter et al., 2012). Brocca et al. (2010) and Vachaud et al. (1985) demonstrated a strong temporal stability of soil moisture variability, indicating a persistent soil moisture pattern over time, which is the main cause of the observed systematic error. Moreover, the dielectric properties of the substrate, influenced by soil properties such as clay content, soil organic matter, bulk density, and soil salinity, may affect the soil moisture measurements of dielectric sensors. Within a MZ, mainly microscale soil moisture variability, resulting from variations in soil particle and pore size, preferential flow, plant roots, microtopography, and localized irrigation practices (e.g., drip irrigation), may significantly impact soil moisture sensor measurements depending on the exact position of the sensor. Finally, systematic errors may also arise due to incorrect sensor installation, e.g., when sensors are installed too deep or inserted vertically instead of horizontally or are influenced by air gaps, their measurements may be consistently biased. On top of an additive bias, such improper installation could also lead to a multiplicative systematic error, which was not considered in this study but is included in the random error term.

The difference between the systematic error of individual sensors and the systematic error of averaged sensor measurements was smaller than what would be expected if the time-persistent deviations between the individual sensor measurements and the true SWC would be independent between different sensors. This could be due to spatial correlation of soil moisture that exists within the range of distances between the different sensors in a MZ. As such, sensors that are close together, i.e., sensors that are spatially correlated, will have similar systematic deviations from the true SWC in the MZ due to similar soil and plant properties. Spatial sensor correlation will be further discussed in Sect. 5.3.

### 5.2.2 Random errors

After estimating the sensor-specific and MZ-specific intercepts, the random errors were quantified for both the individual sensors and the averaged sensor measurements relative to their respective curves (Table 2). The pooled random error variance of the individual sensors ($\sigma_\epsilon^2$) was quantified based on the sensor measurement deviations with respect to their sensor-specific curve using Eq. (8), and resulted in a standard deviation of 0.034 m³ m⁻³ ($\sigma_\epsilon^2 = 0.001183$). The random error variance of the individual sensors was divided by three to obtain the random error variance of the MZ-averages under the assumption of sensor independence (Eq. (16)), which resulted in $\sigma_{\bar{\epsilon}}{}^2 = 0.000394$. Then, the pooled random error variance of the MZ-averages was quantified based on the sensor measurement deviations with respect to their MZ-specific curve using Eq. (13), which resulted in a standard deviation of 0.032 m³ m⁻³ ($\sigma_{\bar{\epsilon}}{}^2 = 0.000998$) that was considerably larger than the estimate based on the individual sensors assuming non-correlated random errors.

Fluctuations in environmental conditions, vertical soil moisture (re)distribution and measurement timing affect all sensors equally, resulting in correlated temporal errors across all sensors within a MZ. This process-related sensor correlation will be further discussed in Sect. 5.3.

### 5.2.3 Total error variance and error autocorrelation

Finally, the total error variance and error autocorrelation were quantified for both the individual sensors and the averaged sensor measurements (Table 2). The pooled total error variance of individual sensor measurements was 0.002563 ($\sigma_{tot} = 0.051$ m$^3$ m$^{-3}$), and the error autocorrelation (AR) was 0.538. The total error variance of the field averages derived under the assumption of sensor independence using Eq. (17) resulted in $\frac{\sigma_{tot}^2}{n} = 0.000854$. The pooled total error variance of the average measured SWC using Eq. (14) was $\sigma_{\overline{tot}}^2 = 0.002068$ ($\sigma_{\overline{tot}} = 0.045$ m$^3$ m$^{-3}$). The pooled systematic error variance $\sigma_{\overline{\alpha}}^2$ was similar to the pooled random error variance $\sigma_{\overline{\epsilon}}^2$, which caused a strong error autocorrelation of 0.518.

Even though $\sigma_{\overline{tot}}^2$ was significantly different from $\frac{\sigma_{tot}^2}{n}$, the difference between the AR values was negligibly small, which implies that the random temporal error and the time-invariant systematic error were affected similarly by sensor dependency. The process-related deviations that are temporally varying but correlated among sensors, $\beta$, are also affected by spatial dependency, i.e., sensors placed in close proximity are more likely to be at locations with similar soil hydrological process-behavior, characterized by similar soil parameters such as porosity and hydraulic conductivity.

The pooled error autocorrelation found in this study was lower than expected based on previous studies (Hendrickx et al., 2023). A larger spatial soil variability within a MZ would generate a larger systematic deviation of an individual sensor or a group of sensors installed at a specific location in the MZ from the average soil moisture in the MZ. This may be the result of a larger inherent soil heterogeneity, or a larger MZ area. If the random error remains stable while the MZ area expands, the error autocorrelation is likely to increase. In contrast, error autocorrelation of the average sensor measurement is expected to decrease with decreasing spatial sensor correlation. Hence, as sensors are located further apart, spatial sensor correlation decreases resulting in a decrease in error autocorrelation. If the sensors are not biased inherently but their bias is position-dependent as was the case here, the bias of the average sensor measurement will decrease with an increasing number and broader coverage of sensors, again resulting in a decrease in error autocorrelation. The previous study of Hendrickx et al. (2023) assessed error autocorrelations of the deviations of an individual sensor compared to the average sensor measurement in the field, which was much larger than a MZ in this study, and found error autocorrelations close to 1, as could be expected due to the larger systematic deviations at field scale compared to MZ scale and smaller random errors due to a one-on-one comparison of sensor time series instead of soil moisture samples with a different measurement volume.

The standard error of the soil moisture sample mean of nine subsamples ($\sigma_{samp} = 0.0038$ m$^3$ m$^{-3}$) was 12 times smaller than the total error standard deviation of a mean sensor measurement from three sensors ($\sigma_{\overline{tot}} = 0.045$ m$^3$ m$^{-3}$, Table 2), which would result in a much larger weight of the soil moisture samples in a data assimilation context.

**Table 2 Summary of error variances and standard deviations derived from individual and averaged sensor measurements, with a number of $N = 3\times375$ data points.**

| | Individual sensor (Eqs. (6)-(10)) | MZ-averaged assuming spatial sensor independence (Eqs. (15)-(17)) | MZ-averaged (Eqs. (11)-(14)) |
|---|---|---|---|
| **Number of intercepts** | 279 | 279 | 93 |
| **Systematic error $\sigma_\alpha^2$** ($\sigma_\alpha$) | 0.001380 (0.037 m$^3$ m$^{-3}$) | 0.000460 (0.021 m$^3$ m$^{-3}$) | 0.001070 (0.033 m$^3$ m$^{-3}$) |
| **Random error $\sigma_\epsilon^2$** ($\sigma_\epsilon$) | 0.001183 (0.034 m$^3$ m$^{-3}$) | 0.000394 (0.020 m$^3$ m$^{-3}$) | 0.000998 (0.032 m$^3$ m$^{-3}$) |
| **Total error $\sigma_{tot}^2$** ($\sigma_{tot}$) | 0.002563 (0.051 m$^3$ m$^{-3}$) | 0.000854 (0.029 m$^3$ m$^{-3}$) | 0.002068 (0.045 m$^3$ m$^{-3}$) |
| **AR** | 0.538 | 0.538 | 0.518 |
| $\sigma_\beta^2$ (Eq. (23)) | 0.000905 (0.030 m$^3$ m$^{-3}$) | NA | NA |
| $\sigma_{\epsilon,nc}^2$ | 0.000279 (0.017 m$^3$ m$^{-3}$) | NA | NA |

## 5.3 Spatial sensor correlation assessment

The degree of temporally stable spatial correlation was assessed by performing numerical calculations and spatial analysis on the systematic errors. As the variance of the sensor-specific intercepts ($\sigma_\alpha^2$) was significantly smaller than three times the variance of the MZ-specific intercepts ($\sigma_{\bar{\alpha}}^2$), the sensors could not be considered spatially independent. The correlation coefficient $\rho_\alpha$ was 0.655 (Eq. (19)), indicating strong spatial correlation. Additionally, the intercepts of the three sensors in one MZ showed strong positive correlations with an average Pearson correlation of 66.5% (Appendix A: Fig. A3). The construction and assessment of a small-scale semivariogram can be found in Supplementary Materials (S2).

The process-related sensor correlation was quantified by comparing the random error variance of the individual sensors ($\sigma_\epsilon^2$) and the random error variance based on the average of the three sensors ($\sigma_{\bar{\epsilon}}^2$). If the random errors of the sensors were independent, $\frac{\sigma_\epsilon^2}{n}$ would be equal to $\sigma_{\bar{\epsilon}}^2$, which was not the case. The correlation coefficient $\rho_\epsilon$ was 0.765 (Eq. (23)), which resulted in $\sigma_\beta^2 = 0.000905$ ($\sigma_\beta = 0.030$ m$^3$ m$^{-3}$). This means that only 24% of the total 'random' error was sensor-independent, and assuming sensor independence would result in inaccurate error estimates.

## 5.4 Assumptions

First of all, the pooled error model approach assumes **linearity** between the true soil moisture contents in the top 30 cm soil layer in a 80 m² MZ and the soil moisture contents derived from the sensor measurements at 15 cm depth with the manufacturer's calibration function. The (perpendicular) residual plot of the pooled linear sensor calibration shows randomly scattered residuals (Appendix A: Fig. A1), while the composite soil moisture samples, representing the true soil moisture, and the mean sensor measurements showed a high correlation of 83%, both suggesting that the linearity assumption is valid. Additionally, second-, third- and fourth-degree polynomial regression models were compared with a linear regression model using the Akaike Information Criterion (AIC), which suggested that the linear model would be the most appropriate choice (Appendix A: Fig. A4).

Secondly, the pooled error model approach assumes **error stationarity and error orthogonality**, i.e., error variances do not change over time, and are therefore also independent of the soil moisture state. Previous studies have, however, shown that spatial soil moisture variability is dependent on the mean soil moisture state (Albertson and Montaldo, 2003; Famiglietti et al., 2008; Hendrickx et al., 2023; Lawrence and Hornberger, 2007; Manns et al., 2014; Pan and Peters-Lidard, 2008; Rosenbaum et al., 2012; Schlüter et al., 2013; Teuling and Troch, 2005; Vereecken et al., 2007). Nonetheless, we argue that these

assumptions are acceptable during the growing season in an irrigated field as the temporal soil moisture range in such a field will likely be narrow. If the error model were to consider a multiplicative systematic error, the errors would depend on the soil moisture state, i.e., errors would be non-orthogonal. Furthermore, dynamic errors that are independent from soil moisture state might occur when soil properties such as bulk density and field capacity change over time (Jirků et al., 2013).

Additionally, the pooled error model approach assumes **spatial consistency**, i.e., the pooled total error variance ($\sigma_{\overline{\text{tot}}}^2$) and the pooled error covariance ($\sigma_{\overline{\alpha}}^2$) are considered to apply for all fields and all soils in the area of Flanders in which data were collected in a set of fields well-spread over that area, which implies that we regard the MZs in the different fields over the different years as having equal spatial soil moisture variability. We argue that this assumption is acceptable for MZs of about 80 m², which is smaller than the scale over which significant variations in soil texture, soil properties, topography etc. may occur within a field, and that the effects of these factors may be reduced due to uniform vegetation in flat, irrigated fields where extreme drying and large temporal soil moisture fluctuations are minimized. A zone of 80 m² in such irrigated, tilled agricultural fields is small compared to correlation lengths of field-scale soil moisture variability of 10 to 70 meters as described in literature (De Lannoy et al., 2006; Vereecken et al., 2014). Therefore, the variability that we consider is related to microscale variations. Whether the results are also applicable to grasslands or no-till fields would be an interesting future research topic. Moreover, the irrigation method applied during a cropping cycle may impact systematic deviations of the sensors compared to the composite soil moisture samples. As discussed in Supplementary Materials (S3), a pooled error model for a specific irrigation method could result in a more accurate error estimation, but would require an extensive dataset for that specific irrigation method.

Our motivation to assume a general soil moisture variability and covariance, i.e., assuming temporal and spatial independence, is to use it as a best "guesstimate" which can be used in data assimilation or inverse modeling to estimate model parameter and model prediction uncertainty. Estimating the variance and temporal covariance of soil moisture measurements in a specific field or MZ would require a large number of sensors. Then, an accurate estimate of the variance and covariance is obtained but at the same time also an accurate estimate of the soil moisture is obtained. The paradox is that when soil moisture variability can be estimated accurately, its estimate may not be so relevant anymore since then, the mean soil moisture will also be estimated accurately and model parameter uncertainty and prediction uncertainty will depend more on the model error rather than on the errors or uncertainties in the measurements. Further studies are required to evaluate the impact of the assumptions that we make on the resulting parameter and model prediction uncertainty, as well as the impact of a violation of the assumptions compared to not having an estimate of the error variance and covariance at all.

When deriving the errors of the MZ-averages from the errors of individual sensors (Eq. (15)-(16)), **sensor independence** is typically assumed. The process-related sensor correlation (impacting random errors) and the spatial correlation (impacting systematic errors) found in this study were 76% and 66%, respectively (Sect. 5.3). The strong correlations that were observed suggest that assuming sensor independence would be incorrect, and the errors of the MZ averages should be computed directly based on the average measurements using Eqs. (11)-(14). The impact of spatial correlation on observational errors could be reduced by obtaining larger spatial coverage, e.g., by installing more sensors and by positioning the sensors further apart. Reducing the spatial correlation is however not required as long as it is accounted for in the error model and error covariance matrix.

Furthermore, the pooled error model approach assumes **zero cross-correlation** between the errors of the soil moisture samples and the observational errors of the sensor measurements. This assumption was satisfied as the individual soil moisture samples were taken within a radius of 5 m around the sensors and at different locations each time, and they sample different soil volumes. Additionally, it is important to note that the sensor data in a specific MZ are not calibrated with the MZ-specific samples, but instead are calibrated with the pooled sensor calibration which is based on all MZs, ensuring this observational error independence.

Finally, the error models (Eq. (6) and (11)) assume **normally distributed errors**. A normal QQ plot with acceptable heavy tails supports the assumption of normally distributed errors (Appendix A: Fig. A5), while the distributions of the sensor-specific intercepts substantiate the normal distribution of the systematic errors (Appendix A: Fig. A3).

## 6 Application in Bayesian inverse modeling: A case study

We present an illustrative case study of a soil water balance parameter estimation in which the pooled error covariance matrix

is used in a Bayesian inverse modeling algorithm. The model is a single-layered soil water balance "bucket" model, based on FAO-56 approaches (Allen et al., 1998), that computes a daily soil water balance for the growing root zone using weather data, soil and crop parameters as input, and applies a dual crop coefficient approach. The parameter estimation was performed using the Differential Evolution Adaptive Metropolis algorithm (DREAM$_{(ZS)}$), which efficiently explores the parameter space through a multi-chain Markov Chain Monte Carlo (MCMC) approach. This method ensures robust uncertainty quantification

and convergence to the posterior distribution of the estimated parameters, given the soil moisture observations. The algorithm applies the loglikelihood function that includes the full error covariance matrix (Eq. (24)) as objective function.

$$\mathcal{L}(\mathbf{x}; \boldsymbol{\mu}, \boldsymbol{\Sigma}) = -\frac{n}{2}\ln(2\pi) - \frac{1}{2}\ln(|\boldsymbol{\Sigma}|) - \frac{1}{2}(\mathbf{x} - \boldsymbol{\mu})^{\mathrm{T}}\boldsymbol{\Sigma}^{-1}(\mathbf{x} - \boldsymbol{\mu}) , \qquad (24)$$

where $\mathbf{x}$ is a vector with the model simulations, $\boldsymbol{\mu}$ is a vector with the mean observations and $\boldsymbol{\Sigma}$ is their corresponding error covariance matrix.

Twelve uncertain physical model parameters were estimated in DREAM$_{(ZS)}$ using soil moisture sensor measurements and

555 composite soil moisture samples of nine individual gouge auger samples, identical to the measurement setup described in Sect. 2. The parameters that are estimated include the crop factors during the initial, mid-season and end stage ($K_{cb,ini}$, $K_{cb,mid}$ and $K_{cb,end}$, respectively), as well as the lengths of the initial, development and mid-season stage ($L_{ini}$, $L_{dev}$ and $L_{mid}$, respectively). Additionally, the soil moisture content at field capacity ($\theta_{FC}$), the logarithm of the saturated hydraulic conductivity ($\ln(K_{sat})$), the curve number (CN) for runoff estimation, the maximum groundwater table depth ($z_{GWT,max}$), the maximum root depth

($z_{r,max}$) and the initial soil moisture content ($\theta_{ini}$) are estimated.

This case study focuses on the soil water balance of an irrigated chicory field in Herent in 2023. The pooled error covariance matrix is applied and results are compared to the parameter estimation where error covariance is assumed to be zero to highlight its importance. The sensor measurements are calibrated using the pooled sensor calibration, and the remaining bias between soil samples and mean sensor data is demonstrated in this case study.

Since the standard error of the composite soil moisture sample was 12 times smaller than the total error standard deviation of a mean sensor measurement, and a soil sample was available every two weeks in contrast to the daily sensor measurements, the weight of the sensor data and soil samples in the parameter estimation was similar on average over the whole cropping cycle. However, the presence of sensor error autocovariance affects how these weights influence the parameter estimates. In Fig. 8a, the predicted SWC is shown with its 95% confidence interval (CI) for the configuration in which both sensor data and

soil samples are used in the DREAM$_{(ZS)}$ parameter estimation while applying the pooled error covariance matrix. Due to the high sensor error autocorrelation of 0.518 (Table 2), the framework is able to correct for the bias in the sensor data.

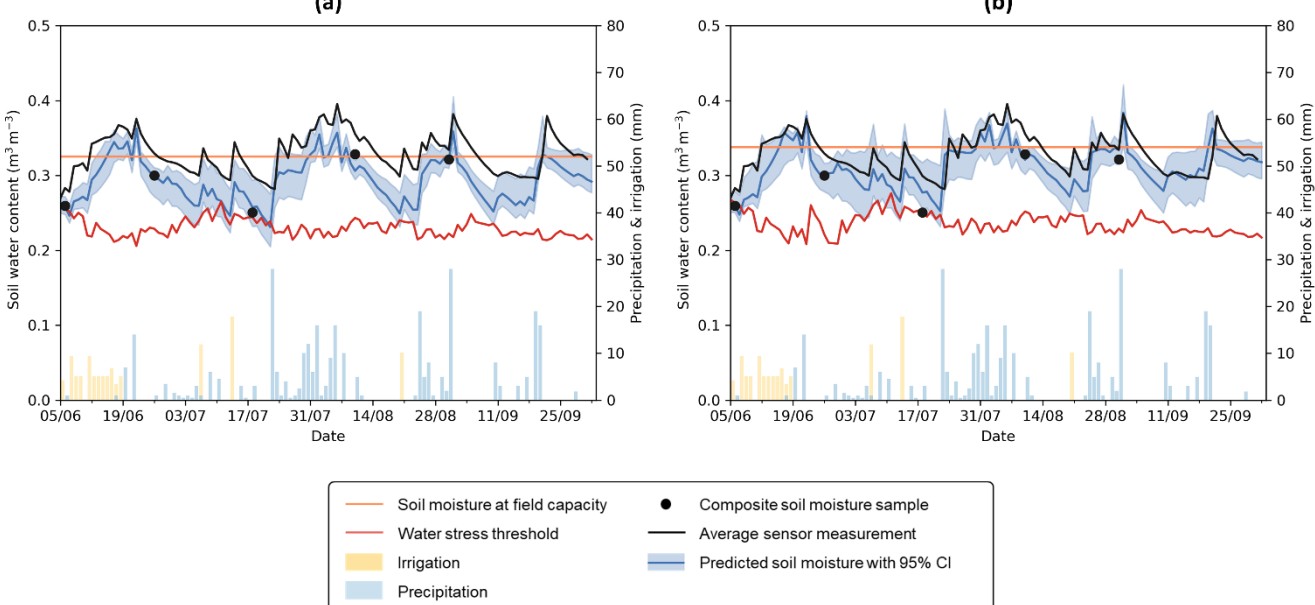

**Fig. 8 Predicted SWC with a 95% CI (blue) after parameter estimation in DREAM(zs) using daily, averaged soil moisture sensor measurements (black) and composite soil moisture samples (●) while applying the pooled error covariance matrix (a) and while assuming zero error covariance (b).**

In contrast, Fig. 8b shows the predicted SWC with its 95% CI under the assumption of zero observational sensor error covariance. Although the total variance of the observational errors remained unchanged, the model uncertainty is generally larger than in Fig. 8a. Since all error variability is now assumed to be random, soil moisture dynamics are not captured well in the model calibration (e.g., 28/06 – 25/07). Additionally, the framework does not correct for the systematic bias in the sensor data, which is most visible during periods without a soil sample (e.g., 11/09 – 1/10).

## 7    General discussion and Conclusions

In this study, a pooled error approach was presented and illustrated using soil moisture measurement data from 93 cropping cycles in agricultural fields across Flanders, which were used to quantify the pooled error variance, error covariance, and error autocorrelation of daily soil moisture sensor data. The pooled results apply to a sensor setup with three TEROS 10 sensors that are located close together within a MZ of about 80 m², and apply to fields in the area of Flanders with soil textures ranging from sand to silt loam. The case study presented in Sect. 6 serves as a preview for future work, and illustrates the implementation of the pooled error covariance matrix in a Bayesian inverse modeling framework.

While this paper focusses on error modeling in soil moisture sensing, the proposed observational error modeling approach is applicable in various contexts characterized by (1) spatial heterogeneity impacting point measurements, (2) continuous measurements at fixed locations with few repetitions and limited spatial extent due to practical constraints and cost considerations, and (3) reference measurements representing the true average of a MZ. Such measurement setups are prevalent across diverse domains and applications, including but not limited to environmental monitoring, (agro)geophysical monitoring, water quality monitoring, and agricultural management. For example, a stem water potential sensor (e.g., FloraPulse) can be considered the equivalent of a soil moisture sensor, providing automated high-frequency readings but being limited to only a few sensors. Manual stem water potential measurements are the equivalent of manual soil moisture sampling, where measurements on a sufficient number of leaves in the plot can provide a reliable average, but the process is too labor-intensive to perform frequently. Similarly, sap flow sensors, which provide high-frequency data but are expensive and show variability between trees and locations on a tree stem, need to be calibrated with independent observations of transpiration. This calibration can be achieved through longer-term observations of water balance components over several weeks for a MZ,

yielding only a few data points over time. In water quality monitoring, multi-parameter sensors for surface water, which are expensive but provide high-frequency data, must be complemented with manual water sampling and laboratory analysis. In agrogeophysical monitoring, methods such as electrical resistivity tomography (ERT) or electromagnetic induction (EMI) are often used to map soil moisture variability across agricultural fields. These methods provide spatially extensive snapshots of subsurface conditions, but they typically require calibration with point measurements, such as soil moisture sensors, to improve

accuracy. The pooled sensor calibration and the MZ-specific systematic deviations between the calibrated sensor data and (unbiased) sampling data found in this study underline once more that it is essential to consider potential biases of the point-based sensor measurements relative to the true values when using sparse sensor networks to calibrate these geophysical measurements. Similar to the discrepancy between the soil moisture sample of the 30 cm layer and the point measurement of the sensor at 15 cm depth, the point measurement might not be representative for the measurement resolution (both vertical

and horizontal) of the geophysical measurement.

The proposed pooled error approach initially requires an extensive measurement dataset, but minimizes MZ-specific measurement requirements. This approach could also be applied to identify observational errors at larger scales, such as management zone or field level. This means that, rather than requiring an extensive sensor network in each field or management zone, only few sensors are required in combination with temporally sparse reference measurements. Even with practical and

budgetary limitations, this approach permits to estimate soil moisture observational errors, both systematic and random, and to correctly allocate weight and confidence to different types of measurements. For example, in Bayesian model state and parameter estimation, independent uncorrelated soil moisture samples with a low uncertainty may have a significant impact on model bias reduction when sensor observational errors are highly autocorrelated, as was illustrated by the case study. A limitation of this approach is the assumption that all MZs have equal local heterogeneity, which might not always hold true.

However, overcoming this limitation would require a denser measurement network in each field, which is exactly what we are trying to avoid. Furthermore, geophysical methods such as ERT and EMI can be applied to quantify spatial variability in a field, but this is typically done only once per crop cycle, often at the start before the crop is established. Integrating these spatial geophysical data with in-situ point measurements can enhance the robustness of parameter estimation in soil hydrological models by providing higher data accuracy and appropriate error propagation. Such joint datasets offer the

advantage of capturing both spatial and temporal variations in soil moisture, which are critical for effective irrigation management. This also raises the question of whether a sparse sensor network within one management zone can be used to extrapolate dynamics to other management zones, or if a MZ is required in each management zone. Additionally, it is important to note that sensor measurements in different MZs or management zones within a field could exhibit different spatial patterns compared to the actual conditions and geophysical observations due to varying biases ($\alpha$) between sensors and ground-truth

(e.g., see Fig. 6).

In the context of Bayesian model state or parameter estimation, the accuracy of the error estimates will also significantly impact optimization results, including the uncertainty estimate on the model state or parameter, as inaccurate error estimates will propagate through Bayesian inference. Additionally, adoption of the pooled error approach yields an error covariance matrix that is invertible and well-conditioned. This computational attribute holds particular significance in likelihood

computation procedures, ensuring numerical stability and facilitating accurate statistical inference.

To conclude, the pooled error modeling approach facilitates low-density in situ sensor measurement networks while still being able to estimate soil moisture variability and error autocorrelation by assessing MZ-specific biases and random errors. This approach is particularly relevant for agrogeophysical studies, where understanding soil moisture dynamics and their uncertainty is critical for decision-making in agriculture. Neglection of error autocorrelation is a common but incorrect

assumption when measurements have limited spatial coverage, as was illustrated by the substantial AR of 0.518 found in this study. Future research is needed to evaluate the impact of this pooled error model and uncorrelated soil moisture samples on

parameter estimation in Bayesian soil hydrological modeling. Additional work is required to test if the results, i.e., the error variances and autocorrelation found in this study, are applicable in other regions, for different land uses, or alternative setups.

**Acknowledgements.** This work was funded by the Research Foundation Flanders (FWO fellowship 1S20822N). The study sites were monitored in the context of a project funded by Flanders innovation & entrepreneurship (VLAIO project HBC.2018.2201). We acknowledge the support of the Soil Service of Belgium, who collected the field data, as well as the research centers in Flanders as partners (*Provinciaal Proefcentrum voor de Groenteteelt Oost-Vlaanderen* (PCG) – now Viaverda – , *Proefstation voor de Groenteteelt* (PSKW), *Praktijkpunt Landbouw Vlaams-Brabant*). We sincerely thank the

anonymous reviewers for their valuable comments and suggestions, which have helped improve the quality of this manuscript.

**AI statement:** In the preparation of this manuscript, AI tools were used for enhancing the readability and quality of the text.

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

**Appendix A: Supplementary figures**

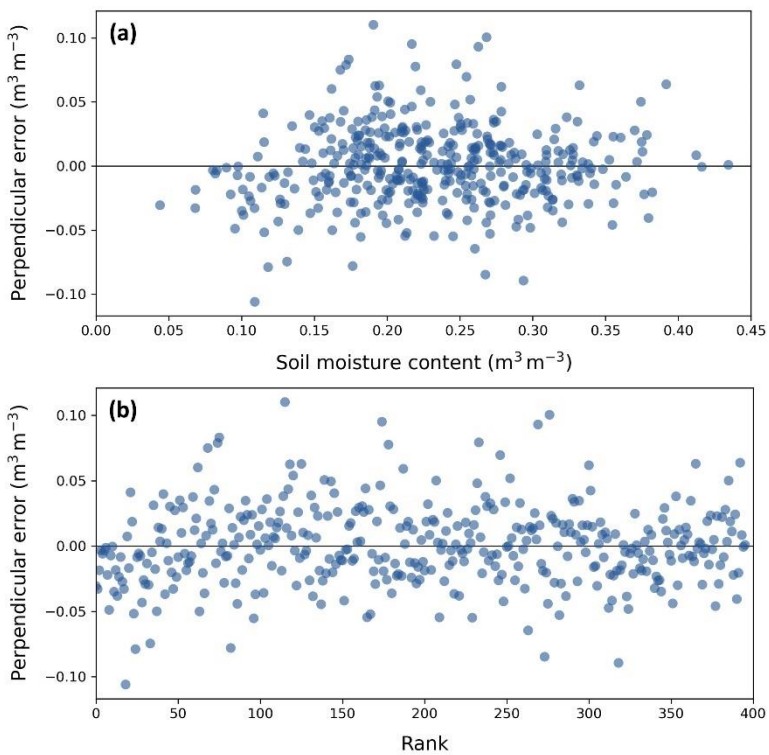

Fig. A1 Perpendicular residuals after applying the orthogonal Deming regression to the sensor data (Eq. (3), Fig. 2), as a function
of SWC (a), and as a function of SWC rank (b).

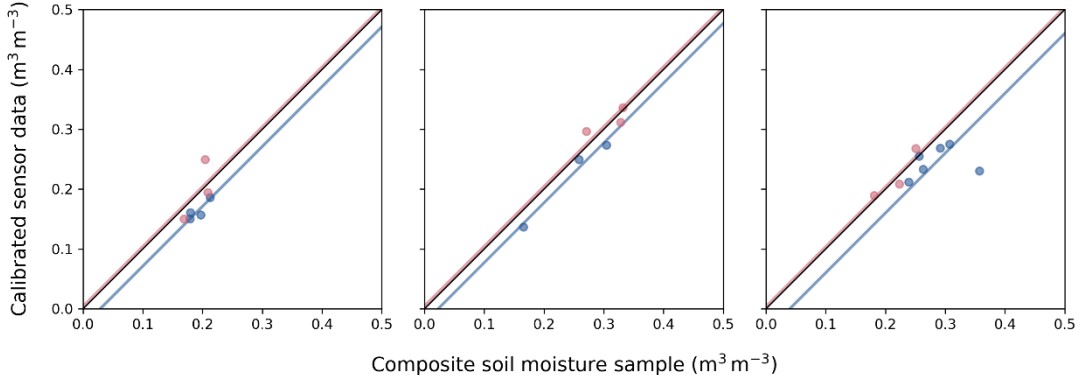

Fig. A2 Examples of double cropping cycles on a certain field within one year that show how the mean bias (intercept) shifts after
the sensors are removed and reinstalled (blue: first cropping cycle, pink: second cropping cycle).

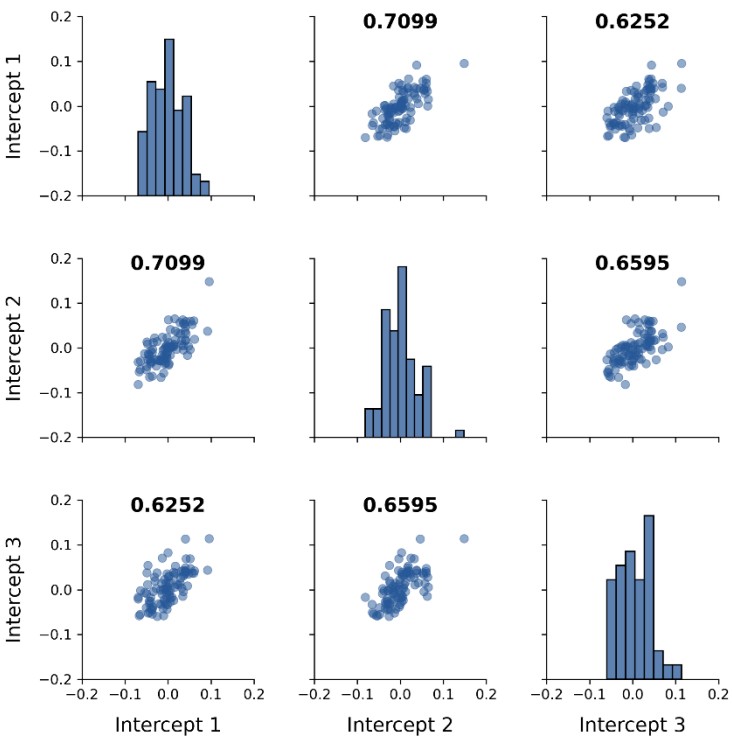

**Fig. A3** Pair plot of the three sensor-specific intercepts of each cropping cycle with their Pearson correlations indicated in bold.


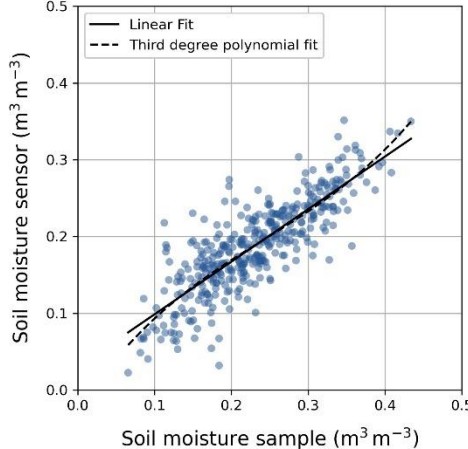

**Fig. A4 If we would not opt for an orthogonal Deming regression, but instead fit a model with the soil moisture sensor data being the uncertain dependent variable ($y$) and the soil moisture sample data (ground-truth) being the independent variable ($x$), we can compare different models using the Akaike Information Criterion (AIC). To determine whether a higher-degree polynomial fit would be more appropriate, we tested a second-, third- and fourth-degree polynomial model. The best polynomial model was a third-degree fit with AIC = -1571.6, while the linear model was similar and even slightly better with AIC = -1572.3. The linear regression model and the third-degree polynomial model are plotted.**


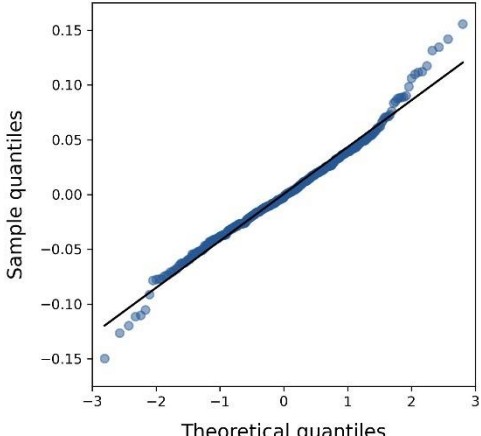

**Fig. A5 Normal QQ plot of error residuals**

**Appendix B: Statistical definition on the autocovariance of observational sensor errors equal to $\sigma_\alpha^2$**

We can show that the autocovariance of observational errors of sensor measurements is equal to the systematic error variance $\sigma_\alpha^2$. We start from our error model formulation in Eq. (6): $E_{i,k} = \alpha_k + \epsilon_{i,k}$, where $\alpha_k$ is the systematic error for sensor $k$, and $\epsilon_{i,k}$ is the random error for measurement $i$ for sensor $k$. The random error is assumed to have zero mean and to be uncorrelated so that $\text{Cov}(\epsilon_{i,k}, \epsilon_{j,k}) = 0$ for $i \neq j$. The systematic error variance is $\sigma_\alpha^2 = \text{Var}(\alpha_k)$.

The autocovariance for two errors $E_{i,k}$ and $E_{j,k}$ within the timeseries of sensor $k$ ($i \neq j$) is then defined as:

$$\text{Cov}(E_{i,k}, E_{j,k}) = \text{Cov}(\alpha_k + \epsilon_{i,k}, \alpha_k + \epsilon_{j,k}) . \tag{B1}$$

Using the linearity of covariance, this expands to:

$$\text{Cov}(E_{i,k}, E_{j,k}) = \text{Cov}(\alpha_k, \alpha_k) + \text{Cov}(\alpha_k, \epsilon_{j,k}) + \text{Cov}(\epsilon_{i,k}, \alpha_k) + \text{Cov}(\epsilon_{i,k}, \epsilon_{j,k}) . \tag{B2}$$

We can simplify this notation. The first term is simply the variance of $\alpha$:

$$\text{Cov}(\alpha_k, \alpha_k) = \sigma_\alpha^2 . \tag{B3}$$

Since $\alpha_k$ is independent of the random error $\epsilon_{.,k}$, these covariances are zero:

$$\text{Cov}(\alpha_k, \epsilon_{j,k}) = \text{Cov}(\epsilon_{i,k}, \alpha_k) = 0 . \tag{B4}$$

Since $\epsilon_{i,k}$ and $\epsilon_{j,k}$ are uncorrelated for $i \neq j$, this term is also zero:

$$\text{Cov}(\epsilon_{i,k}, \epsilon_{j,k}) = 0 . \tag{B5}$$

Thus, the autocovariance of observational errors for measurements within a timeseries of a sensor $k$ is equal to the systematic

error variance:

$$\text{Cov}(E_{i,k}, E_{j,k}) = \sigma_\alpha^2 . \tag{B6}$$

**Appendix C: Derivation of variance of the mean (Eqs. (15)-(16)) and spatial correlation (Eqs. (19) and (23))**

For $n$ random variables $(X_1, X_2, \dots, X_n)$, the variance of their average is given by Eq. (C1).

$$\text{Var}\left(\frac{X_1 + X_2 + \dots + X_n}{n}\right) = \frac{1}{n^2} \text{Var}(X_1 + X_2 + \dots + X_n) \, . \tag{C1}$$

If the covariances between these variables are equal, i.e., $\text{Cov}_{XX} = \text{Cov}(X_1, X_2) = \dots = \text{Cov}(X_{n-1}, X_n)$, the variance of the sum of the variables is given by Eq. (C2).

$$\text{Var}(X_1 + X_2 + \dots + X_n) = \text{Var}(X_1) + \text{Var}(X_2) + \dots + \text{Var}(X_n) + (n - 1)n\text{Cov}_{XX} \, . \tag{C2}$$

If the variances of these variables are also equal, i.e., $\sigma^2 = \text{Var}(X_1) = \text{Var}(X_2) = \dots = \text{Var}(X_n)$, the variance of the average $(\sigma_{\text{mean}}^2)$ can be written as Eq. (C3).

$$\sigma_{\text{mean}}^2 = \frac{1}{n^2}\left[n\sigma^2 + (n - 1)n\rho\sigma^2\right], \tag{C3}$$

where $\rho$ is the correlation, defined as $\frac{\text{Cov}_{XX}}{\sigma^2}$. When both the variance of the $n$ individual variables $(\sigma^2)$ and the variance of their average $(\sigma_{\text{mean}}^2)$ are known, the correlation can be quantified using Eq. (C4).

$$\rho = \frac{n\sigma_{\text{mean}}^2 - \sigma^2}{(n-1)\sigma^2} \, . \tag{C4}$$

The correlation between measurements of three sensors $(n = 3)$ can then be quantified using Eq. (C5).

$$\rho = \frac{3\sigma_{\text{mean}}^2 - \sigma^2}{2\sigma^2} \, . \tag{C5}$$

**Code availability**

Python scripts are available upon request.

**Data availability**

The dataset generated and used in this study is available at KU Leuven RDR: https://doi.org/10.48804/M6WKTN under a CC-BY 4.0 license.

**Author contribution**

All co-authors contributed to the design of the field measurement trials, which were carried out by the research centers, led by Pieter Janssens, Sander Bombeke, Evi Matthyssen, and Anne Waverijn, who primarily contributed through resource provision, collaboration with farmers, and investigation. Pieter Janssens was responsible for project administration and supervision. Marit Hendrickx developed the model code, performed the computations and simulations, and conducted the conceptualization and methodology for error modeling, with contributions from Jan Vanderborght, Pieter Janssens and Jan Diels. Marit Hendrickx was responsible for data curation, formal analysis, and visualization. The manuscript was prepared by Marit Hendrickx, with contributions mainly from Jan Vanderborght, Pieter Janssens, and Jan Diels.

**Competing interests**

The authors declare that they have no conflict of interest.
Some authors are members of the editorial board of journal SOIL.

**Short summary**

We developed a method to estimate errors in soil moisture measurements using limited sensors and infrequent sampling. By analyzing data from 93 cropping cycles in agricultural fields in Belgium, we identified both systematic and random errors for our sensor setup. This approach reduces the need for extensive sensor networks and is applicable to agricultural and environmental monitoring, and ensures more reliable soil moisture data, enhancing water management and improving model predictions.