# Peer review of "Pooled Error Variance and Covariance Estimation of Sparse In Situ Soil Moisture Sensor Measurements in Agricultural Fields in Flanders"

_EGUsphere, 2024_

## Author Comment (AC2)

[revised manuscript text omitted]

---

## Author Response (AR1)

**AUTHOR RESPONSE**

**REVIEWER 1**

We would like to thank this reviewer for their time and effort in reviewing our manuscript. Even though this reviewer indicated not having the optimal background, the evaluation is thorough and constructive, which will greatly help us improve the quality and clarity of our work. We highly appreciate the kind words on the experimental work and the relevance of this work.

Our responses are organized according to the reviewers comments, first repeating the comment after which we state our answer.

> *"This study presents an approach to estimate error variance and covariance estimates for sparse in situ soil moisture monitoring networks. This is certainly a **relevant topic** that fits well into the scope of the journal and addresses an important issue (i.e., the temporal and spatial error correlation properties of in situ measurements).*
>
> *Unfortunately, I have to admit that I could not follow the method within a reasonable amount of time and mental effort. I am not an expert on in situ soil moisture sampling, so this can certainly be attributed, in part, to a lack of background understanding on my side. Nevertheless, I do think that the manuscript could benefit greatly from **revising the description of the methodology, better motivating why certain steps are taken**, and **backing up certain claims and steps with literature**."*

We understand that the reviewer found the methodology challenging to follow. The description of the methodology will be improved, and specific steps and claims will be better motivated and supported by relevant literature, based on the next comments of this reviewer, as well as the comments and in-text suggestions of Reviewer 2.

> *"For example, I do not understand why, in **Eq. 7** an average of standard deviations is taken, and why the standard error of the mean of standard deviations should represent the measurement error of the composite soil moisture (which is the average of gravimetric measurements, right?)... Why is the error variance of an average standard deviation equal to the error variance of the average of the individual measurements?"*

Thank you for pointing out this concern; Reviewer 2 also requested an explicit assumption statement here. First, the (weighted) average of the standard deviations of the individual soil samples in Eq. 7 represents the overall approximation of the standard deviation of an individual soil sample. In doing so, we assume that there is a fixed common standard deviation of an individual soil sample. This is also why we recommend a multi-sample analysis to obtain a more accurate estimate of the soil moisture sample error for a heterogeneous field or MZ (since our results showed strong homogeneity). We also know that the true measurement variability is probably dependent on the humidity of the soil, which is motivated by previous studies (Hendrickx et al., 2023)[1], but this is a complexity that is not considered in this current study.

Then, the standard error (SE) is calculated to represent the standard deviation of the average of *n* individual soil samples (in this case, the composite soil moisture sample). The reduction in variability due to averaging is reflected in this standard error, as the composite sample inherently has less variability than individual measurements. This approach is consistent with statistical principles (e.g.,
* * *
[1] Hendrickx, M. G. A., Diels, J., Janssens, P., Schlüter, S., and Vanderborght, J.: Temporal covariance of spatial soil moisture variations: A mechanistic error modeling approach, Vadose Zo. J., e20295, https://doi.org/10.1002/VZJ2.20295, 2023.

https://doi.org/10.1136/bmj.331.7521.903 explains the distinction between standard deviations and standard errors). We hope this clarifies the rationale behind the method presented in Eq. 7-8.

Also, as noted by reviewer 2, in line 195, it should say 'pooled variance of composite soil moisture samples' instead of 'pooled _error_ variance […]'.

> ➤ *"Also, I do not get why, in **L226, a "systematic error that is constant over time"** is defined as a random variable... isn't that a self contradiction? Or where does **Eq (14)** come from? Shouldn't a correlation be the ratio between a covariance and the product of two standard deviations? Is this a standard approach, and is there **literature** to back that up?"*

The systematic error ($\alpha_k$) is constant over time for a specific sensor $k$. However, this systematic error is different for each sensor, hence the definition as a random variable from a normal distribution with variance $\sigma_\alpha^2$. This will be clarified in the text as follows: "[...], $\alpha_k \sim \mathcal{N}(0, \sigma_\alpha^2)$ is a systematic error for sensor $k$, and $\epsilon_{i,k} \sim \mathcal{N}(0, \sigma_\epsilon^2)$ is a random error (**Error! Reference source not found.**a). The systematic error is the time-invariant component of the deviation of the sensor measurement $\theta_{\text{sensor},i,k}$ from the true SWC from the sampling, while the random error is the time-variant component."

Moreover, this $\sigma_\alpha^2$ is the autocovariance (i.e., the covariance between errors over time, within the same timeseries) – clarification on this will also be provided in the manuscript as requested by Reviewer 2 (see RC2 response). A correlation is indeed calculated as the ratio of a covariance and the product of two standard deviations:

$$\text{Corr}(X, Y) = \frac{\text{Cov}(X,Y)}{\text{SD}_X \text{SD}_Y} \ .$$

However, here we are calculating an _auto_correlation, meaning the covariance is now the _auto_covariance (equal to $\sigma_\alpha^2$) and the two standard deviations are equal (=standard deviation of the errors), which corresponds then to the total variance of the errors. Using the formula:

$$\text{Corr}(E, E) = \frac{\text{Cov}(E,E)}{\text{SD}_E^2},$$

error autocorrelation is the correlation between errors within the same timeseries, e.g. $E_i$ and $E_j$, and we assumed that it is equal for any pair of measurements $i$ and $j$ in the time series of observations. The standard deviation of each of these errors is also assumed constant over time and equal to $\text{SD}_E$.

> ➤ *"I am also skeptical about the **underlying assumption that (L247) there are no variations** in error variance, covariance, and autocorrelation **between different fields**... How realistic is this assumption? Can you provide a rationale for the belief that this is justified "for MZs of about 80m2"?"*

There are several studies that investigate the dependence of the spatial variance of soil moisture on the heterogeneity of soil properties, topography and vegetation (Albertson and Montaldo, 2003; Hendrickx et al., 2023; Lawrence and Hornberger, 2007; Manns et al., 2014; Pan and Peters-Lidard, 2008; Schlüter et al., 2013; Teuling and Troch, 2005; Vereecken et al., 2007), and the average soil moisture content itself (Bell et al., 1979; Famiglietti et al., 2008; Hendrickx et al., 2023; Irmak et al., 2022; Rosenbaum et al., 2012), and found that all these factors influence soil moisture variability. Assuming a constant soil moisture variability is an assumption that neglects these influences. We argue that the effects of these factors may be reduced in relatively small measurement zones (MZs) with uniform vegetation, in flat, irrigated fields where extreme drying and large temporal soil moisture fluctuations are minimized. A zone of 80 m² in such irrigated, tilled agricultural fields is small compared

[revised manuscript text omitted]

> ➤ *"Lastly, I would recommend to **choose the language in general more carefully**, reducing room for ambiguity or misinterpretations... For example, I belief that statements such as (L280) "measurement errors in a field are correlated with each other", do not actually mean*

> *that the errors of individual measurements are correlated, but that the deviations of the measurements of different sensors, which have been transformed to represent the same soil volume, from soil moisture in that volume are correlated?"*

We understand your concern, and will revise the text to comply to this comment. Reviewer 2 has made similar suggestions throughout the text, which we will also incorporate.

As such, we will define the "measurement error" more clearly, which we are considering to be a lumped error comprised of three sources: two representational errors–one originating from the representation of a whole soil layer by a point sensor measurement (vertical dimension), and the other originating from the representation of soil moisture over a sampled measurement zone of about 80 m² (horizontal dimension)–, and a third 'intrinsic' sensor measurement error which corresponds to the error between the true and the measured soil moisture within the sensor's measurement volume. We will mention and define errors and deviations more clearly in the manuscript, also in the abstract. We will use the term "observational errors", and be sure to differentiate them from "measurement errors", as observational errors include more than the intrinsic error. While these representational and intrinsic errors arise from different causes and dynamics, it is not the aim of this study to distinguish them individually. Instead, we recognize that each of these sources can contribute to both systematic (time-stable) and random (time-variant) errors. This study focuses on accurately quantifying these two components to determine the covariance matrix of the "observational errors".

> ➢ *"Finally, I think the whole **introductory part** around Eqs. (1)-(3) can be omitted because it is **not relevant for the study**. The presented equations are just one choice to integrate the measurements with a model, and whether to use (2) or (3) is merely a choice of ignoring off-diagonals or not... This manuscript does not use the estimated error covariance information in any way, so I recommend simply saying that "it is important to use that in these situations and that's why we try to estimate it", but leave an arbitrarily chosen of how it could (theoretically) be used out of the paper."*

The equations will be omitted from the introduction; instead, few key-references of some of the methods will be mentioned as examples for context.

> ➢ *"In summary: I do believe that the authors have conducted rigorous experiments and are proposing something very relevant and probably sound, I just find myself unable to remove the "probably" from this sentence, because I simply cannot follow the methodology, which, again, could partly be attributed to a lack of understanding on my side, but I recommend nevertheless to revise the methodology and introduction to make the manuscript more accessible to a wider readership."*

We sincerely thank the reviewer for their feedback regarding the clarity of our methodology and introduction. We understand that accessibility to a broader audience is crucial for maximizing the impact of our research. To address this, we will thoroughly revise the methodology and introduction sections to ensure that the key concepts, steps, and rationale are presented in a more clear and accessible manner.

**REVIEWER 2**

We would like to thank this reviewer for their time and effort in reviewing our manuscript. We are pleased to read that the reviewer commends the experimental work and statistical rigor of the analysis. The thorough evaluation, constructive comments, and detailed remarks, questions and suggested edits in the pdf will greatly help us improve the quality and clarity of our work.

Our responses are organized according to the reviewers comments, first repeating the comment after which we state our answer.

> *"This manuscript analyzes the results of a significant soil moisture field measurement campaign in northern Belgium.*
>
> *The motivation of the study is to estimate the **"errors"** that should be assigned to soil moisture measurements when assimilating those data into probabilistic modeling frameworks or using the data in inverse modeling. I put error in quotation marks because **I think that the term is not appropriately defined in this manuscript**. The term as used here includes both 1) the discrepancy between the soil moisture estimate from a sensor and the true soil moisture value in the sensor's measurement volume and 2) the discrepancy between the soil moisture estimate from a sensor and the true mean soil moisture value in some user-defined measurement zone which is far deeper and wider (by orders of magnitude) than the sensor's measurement volume.** These two dimensions of "error" clearly have different physical causes and likely will have different temporal dynamics. Yet here they are lumped together and analyzed as if they are one phenomenon. **I am not sure that this approach is justified**. This is my primary concern with the manuscript."*

Indeed, the "measurement error" we are considering is a lumped error comprised of three sources: two representational errors–one originating from the representation of a whole soil layer by a point sensor measurement (vertical dimension), and the other originating from the representation of soil moisture over a sampled measurement zone of about 80 m² (horizontal dimension)–, and a third 'intrinsic' sensor measurement error which corresponds to the error between the true and the measured soil moisture within the sensor's measurement volume. We will mention and define them more clearly in the manuscript, also in the abstract. We will use the term "observational errors", and be sure to differentiate them from "measurement errors", as observational errors include more than the intrinsic error. While these representational and intrinsic errors arise from different causes and dynamics, it is not the aim of this study to distinguish them individually. Instead, we recognize that each of these sources can contribute to both systematic (time-stable) and random (time-variant) errors. This study focuses on accurately quantifying these two components to determine the covariance matrix of the "observational errors". To justify this lumping approach, we conducted preliminary experiments that are not (yet) discussed in the manuscript:

- At the start of the project, we performed lab tests for TEROS 10 sensors in homogeneous soil columns to test the accuracy and precision of the sensor measurements for repeated readings (~0 m³/m³) and after re-insertion (0.006 m³/m³), and to obtain a lab calibration equation. The resulting calibration curve was [true SWC] = -0.009 + 1.14×[meas.SWC], which is very similar to the pooled on-field calibration curve shown in the manuscript ([true SWC] = -0.006 + 1.26×[meas.SWC]). This lab test will be included in the manuscript as supplementary material.
- Prior to our on-field experiments, we did synthetic experiments with Hydrus-1D to test at which depth the soil moisture content and dynamics best match the soil moisture in the whole 30 cm layer. From these simulations, we found that the 15 cm depth soil moisture best represents the

30 cm layer, which helped us decide on the depth for sensor installation. This could also be included in the manuscript as supplementary material if requested.

> ➢ *"My secondary concern with the manuscript is that the **importance of the results is unclear**. The main outcomes are some estimates of parameters that could be assigned to represent the uncertainty and autocorrelation of the underlying soil moisture sensor data if they were used in data assimilation or inverse modeling. But the data are not used in that way here, so there is **no way to estimate the significance of the results**. The **underlying data appear to be remarkably homogenous, as noted by the authors**. So, in the end, the **statistical rigor of the analysis and the presentation of the work is commendable**, but the **practical importance of the work is unclear**. Clarifying that would improve the value of this contribution."*

With this manuscript, we mainly want to present the approach for pooled/generalized calibration and error modeling when only limited data are available at a single measurement site. The specific outcomes are only applicable to tilled agricultural fields in Flanders with a specific measurement setup with three TEROS 10 sensors. The addition of the lab-based sensor calibration in the next revision will also be an added value for the importance of the manuscript. The similarity between the lab-based sensor calibration and the pooled on-field sensor calibration suggests that this calibration has a broader applicability, e.g. on different fields and in different contexts. Mane et al. (2024)[2] state that general sensor calibrations, i.e., using measurements from multiple sites across a large region, is a viable alternative to field- or soil-specific sensor calibrations, but note that the accuracy is lower. However, our general sensor calibration shows a higher accuracy than the general calibration of the TEROS predecessor, 10 HS (METER), in a study of Vaz et al. (2013)[3], as well as a smaller difference with the lab-based accuracy. These references will also be integrated in the text.

To address the practical importance more fully, our forthcoming manuscript will apply this calibration and error model within a Bayesian soil hydrological model framework to evaluate how sensor errors, autocovariance, and independent samples influence model predictions. This follow-up study, which we expect to submit in early 2025, will explicitly quantify the impact of these uncertainties, providing further insights into the significance and broader implications of this work. This work has already been presented at EGU24 (Hendrickx et al., 2024)[4].

If recommended by the editor, we can add an illustrative case study of a soil water balance parameter estimation in which this pooled error covariance matrix is used in a Bayesian inverse modeling algorithm (DREAM(zs)), and compare it to a zero-covariance assumption case. The specifics can be in Supplementary materials, and the case study may serve as a preview for the upcoming paper.

The reviewer also wondered whether the homogeneous conditions might affect the interpretation of the results and the applicability of the methods in other locations (L344). We acknowledge that the
* * *
[2] Mane, S., Das, N., Singh, G., Cosh, M., and Dong, Y.: Advancements in dielectric soil moisture sensor Calibration: A comprehensive review of methods and techniques, Comput. Electron. Agric., 218, 108686, https://doi.org/10.1016/J.COMPAG.2024.108686, 2024.

[3] Vaz, C. M. P., Jones, S., Meding, M., and Tuller, M.: Evaluation of Standard Calibration Functions for Eight Electromagnetic Soil Moisture Sensors, Vadose Zo. J., 12, vzj2012.0160, https://doi.org/10.2136/VZJ2012.0160, 2013.

[4] Hendrickx, M., Diels, J., Vanderborght, J., Janssens, P. (2024). Field-scale soil moisture predictions using in situ sensor measurements in an inverse modelling framework: SWIM². (Abstract No. EGU24-20013). Presented at the EGU General Assembly 2024, Vienna, Austria, 15 Apr 2024-19 Apr 2024. doi: 10.5194/egusphere-egu24-20013

homogeneous soil moisture conditions observed in this study may limit the generalization of these results. However, the methods remain relevant for exploring sensor errors in more variable environments, as they offer a structured way to quantify uncertainty and autocorrelation, which are critical for applications like data assimilation and inverse modeling.

> ➢ *"I have included further detailed comments, questions, and suggested edits in an attached pdf version of the manuscript."*

The additional comments in the pdf are highly appreciated, and will be handled as such. The pdf with the responses is included as attachment.

To answer for the statistical definition on the autocovariance of sensor measurement errors (**L240**) that was indicated as unclear in the pdf, we can show that the autocovariance of sensor measurement errors is equal to the systematic error variance $\sigma_\alpha^2$. This will also be included in the manuscript.

1. We start from our **error model formulation** (Eq. 10): $E_{i,k} = \alpha_k + \epsilon_{i,k}$,

   where:

   - $\alpha_k$: Systematic error for group (=sensor) $k$.

   - $\epsilon_{i,k}$: Random error for measurement $i$ in group $k$, assumed to have zero mean and to be uncorrelated ($\text{Cov}(\epsilon_{i,k}, \epsilon_{j,k}) = 0$ for $i \neq j$).

2. **Systematic error variance**:

$$\sigma_\alpha^2 = \text{Var}(\alpha_k)$$

3. The **autocovariance** for two errors $E_{i,k}$ and $E_{j,k}$ within the same group $k$ ($i \neq j$):

$$\text{Cov}(E_{i,k}, E_{j,k}) = \text{Cov}(\alpha_k + \epsilon_{i,k}, \alpha_k + \epsilon_{j,k})$$

4. Using the **linearity of covariance**, this expands to:

$$\text{Cov}(E_{i,k}, E_{j,k}) = \text{Cov}(\alpha_k, \alpha_k) + \text{Cov}(\alpha_k, \epsilon_{j,k}) + \text{Cov}(\epsilon_{i,k}, \alpha_k) + \text{Cov}(\epsilon_{i,k}, \epsilon_{j,k})$$

5. We can **simplify** this notation.

The first term is simply the variance of $\alpha$:

$$\text{Cov}(\alpha_k, \alpha_k) = \sigma_\alpha^2$$

Since $\alpha_k$ is independent of the random error $\epsilon_{\cdot,k}$, these covariances are zero:

$$\text{Cov}(\alpha_k, \epsilon_{j,k}) = \text{Cov}(\epsilon_{i,k}, \alpha_k) = 0$$

Since $\epsilon_{i,k}$ and $\epsilon_{j,k}$ are uncorrelated for $i \neq j$, this term is also zero:

$$\text{Cov}(\epsilon_{i,k}, \epsilon_{j,k}) = 0$$

Thus, the autocovariance of sensor measurement errors for measurements within the same group is equal to the systematic error variance:

$$\text{Cov}(E_{i,k}, E_{j,k}) = \sigma_\alpha^2$$

**SUMMARY OF MAIN REVISIONS**

- The abstract has been clarified, with an explicit definition of error now included.
- Throughout the text:
  - The definition of errors is explicitly stated in the introduction.
  - The term "observational errors" is now used instead of "measurement errors," as the latter did not accurately reflect the nature of the errors assessed in this study.
  - The main text has been revised for clarity, with additional explanations and motivations provided where necessary.
  - The language has been refined to be more precise and careful, reducing ambiguity and potential misinterpretations.
- The introductory discussion around Eqs. (1)–(3) has been removed.
- A lab-based sensor calibration has been added to the supplementary materials (S1) and briefly discussed in Section 3.
- A case study (Section 6) has been introduced to illustrate the application of the error covariance matrix.
- Appendix B has been added, providing a statistical definition of the autocovariance of observational sensor errors.